

# A taxonomic review of the genus *Astrocladus* (Echinodermata, Ophiuroidea, Euryalida, Gorgonocephalidae) from Japanese coastal waters

Masanori Okanishi[1], Hisanori Kohtsuka[1] and Toshihiko Fujita[2]

[1] Misaki Marine Biological Station, Graduate School of Science, The University of Tokyo, Miura, Kanagawa, Japan
[2] Department of Zoology, National Museum of Nature and Science, Tsukuba, Tsukuba, Ibaraki, Japan

## ABSTRACT

Japanese species of the genus *Astrocladus* (Echinodermata, Ophiuroidea, Euryalida, Gorgonocephalidae) are reviewed. *Astrocladus coniferus* recently has two junior synonyms, *A. dofleini* Döderlein and *A. pardalis* Döderlein, however, status of these species has long been questioned. These species concepts have not been reviewed in recent years and no molecular phylogenetic analyses have been performed. Observations of the lectotype of *A. coniferus*, as well as the lectotype and four paralectotypes of *A. dofleini* and the holotype of *A. pardalis* have revealed that *A. coniferus* and *A. pardalis* are conspecific and morphologically distinguishable from *A. dofleini*. *Astrocladus coniferus* and *A. dofleini* are supported as distinct species by our molecular data. Additionally, we re-describe *A. exiguus* and *A. annulatus*, based on recently collected specimens and the holotype. We conclude that four species, *A. annulatus*, *A. coniferus*, *A. dofleini*, and *A. exiguus* are present in Japanese waters.

## INTRODUCTION

The gorgonocephalid ophiuroids of the genus *Astrocladus Verrill, 1899* (Ophiuroidea: Euryalida) are characterized by having cone-shaped external ossicles on the disc; a madreporite present on oralmost edge of interradial lateral disc; no calcareous plates on the lateral disc margin; and no arm spines before first branch in adults (*Fell, 1960*; *Baker, 1980*). The genus is distributed from the Indo-Pacific to southern Africa (*Clark & Courtman-Stock, 1976*; *Baker, 1980*; *McKnight, 2000*). It was established by *Verrill (1899)* for *Euryale verrucosum Lamarck, 1816* (= *Astrocladus exiguus* (*Lamarck, 1816*)) and it currently comprises 10 species: *Astrocladus africanus Mortensen, 1933* from southern Africa; *A. annulatus Matsumoto, 1915* from Japan; *A. coniferus* (*Döderlein, 1902*) from Japan; *A. euryale* (*Retzius, 1783*) from southern Africa; *A. exiguus* (*Lamarck, 1816*) from the Indo-West Pacific; *A. goodingi Baker, Okanishi & Pawson, 2018* from the western Indian Ocean; *A. hirtus Mortensen, 1933* from South Africa; *A. ludwigi* (*Döderlein, 1896*)

Corresponding author
Masanori Okanishi, mokanishi@tezuru-mozuru.com

from Indo-Western Pacific; *A. socotrana* *Baker, Okanishi & Pawson, 2018* from northern Indian Ocean, and *A. tonganus* *Döderlein, 1911* from Tonga and New Zealand (*Retzius, 1783*; *Lamarck, 1816*; *Lyman, 1875*, *1882*; *Koehler, 1897*, *1905*, *1907*, *1930*; *Clark, 1911*, *1915*, *1923*; *Döderlein, 1896*, *1911*, *1927*; *Matsumoto, 1915*; *Mortensen, 1933*; *Murakami, 1944a*, *1944b*; *Djakonov, 1954*; *Clark, 1951*, *1974*; *Clark & Rowe, 1971*; *Clark & Courtman-Stock, 1976*; *Irimura, 1969*, *1981*, *1982*; *Cherbonnier & Guille, 1978*; *Baker, 1980*; *McKnight, 1989*, *2000*; *Liao & Clark, 1995*; *Rowe & Gates, 1995*; *Irimura & Tachikawa, 2002*; *Baker, Okanishi & Pawson, 2018*). Of these, three species are currently known from Japanese waters: *A. annulatus*, *A. coniferus*, and *A. exiguus* (*Döderlein, 1911*; *Matsumoto, 1915*; *Irimura, 1981*).

The holotype of *Astrocladus annulatus* was described, from central Japan (*Matsumoto, 1915*, *1917*) and subsequently recorded again from on a specimen from western Japan (*Irimura, 1981*); *Astrocladus exiguus* was recorded from central and western Japan, respectively (*Clark, 1915*; *Irimura, 1981*); and *Astrocladus coniferus* has been recorded from Japanese waters except Hokkaido Island (*Döderlein, 1902*, *1910*, *1911*; *Clark, 1911*; *Matsumoto, 1917*; *Murakami, 1944b*; *Irimura, 1979*, *1981*, *1982*, *1990*; *Yi & Irimura, 1987*; *Fujita & Kohtsuka, 2003*; *Okanishi, O'Hara & Fujita, 2011*; *Kohtsuka et al., 2017*).

In these three species, the taxonomic status of *A. coniferus*, which is currently synonymized with *A. dofleini* and *A. pardalis*, has been controversial. *Astrocladus coniferus* and *A. pardalis* were both originally described by *Döderlein (1902)* in the genus *Astrophyton* and they were mainly distinguished by presence/absence of large conical tubercles on radial shield and dorsal proximal portion of arms (*Döderlein, 1902*). Subsequently, *Döderlein (1910)* moved the two species to *Astrocladus Verrill, 1899* and synonymized *A. pardalis* with *A coniferus*. In the same paper, Döderlein described *Astrocladus dofleini*, which is distinguished from *A. coniferus* in having numerous granular large tubercles on the dorsal disc. Since then, some taxonomists have considered *A. coniferus* (with synonymy of *A. pardalis*) and *A. dofleini* to be separate taxa (*Döderlein, 1910*; *Saba, Tomida & Kimoto, 1982*; *Irimura, 1990*), others have regarded the three taxa as distinct (sub)species (*Matsumoto, 1917*; *Fedotov, 1926*; *Irimura, 1982*; *Shin & Rho, 1996*) or united them as *A. coniferus* without any (sub)specific division (*Murakami, 1944a*, *1944b*; *Fujita & Kohtsuka, 2003*). The type specimens of *A. coniferus*, *A. dofleini* and *A. pardalis* have not been formally re-examined since their original descriptions by *Döderlein (1902*, *1910*, *1911)*.

The other two Japanese species, *A. annulatus* and *A. exiguus*, has been distinguished from *A. coniferus* (and also from *A. dofleini* and *A. pardalis*) by the possession of a covering of external ossicles on the dorsal body. However, no detailed description, including photographs of the body and/or SEM images of ossicles has been provided for *A. annulatus* and *A. exiguus*.

All of the previous studies of *Astrocladus* were based on morphological characters alone. Molecular data for *Astrocladus* spp. has only been published in four articles (*Okanishi et al., 2011*; *Okanishi & Fujita, 2013*; *O'Hara et al., 2017*; *Christodoulou et al., 2019*).

In this study, to address the taxonomic status of *Astrocladus coniferus*, we compared the morphology of the type specimens, for *A. coniferus, A. pardalis* and *A. dofleini*, and

we obtained partial mitochondrial COI genes for seven specimens which were morphologically consistent with type specimens of those three species. Partial mitochondrial COI of the two specimens of *A. exiguus* were also included in the phylogenetic analysis to estimate species delimitation by measuring genetic distance within *Astrocladus*. Additionally, we re-describe *A. annulatus* based on newly collected material, as well as the type specimens, and we re-describe *A. exiguus* based on newly acquired specimens.

## MATERIALS AND METHODS

### Specimens examined

The seven type specimens examined in this study are deposited in The University Museum, The University of Tokyo, Japan (UMUT), the Zoologische Staatssammlung München, Germany (ZSM), and Museum für Naturkunde der Humboldt-Universität zu Berlin, Germany (ZMB). Other newly collected specimens for molecular analysis are deposited in the National Museum of Nature and Science, Tsukuba, Japan (NSMT) (Table 1).

One lectotype of *Astrophyton coniferum* (ZSM 20000897), one lectotype ZSM 20000901/1 and four paralectotypes (three specimens of ZSM 20000901/2, ZSM 20000901/3, ZSM 20000901/3 and one specimen of ZMB 5923) of *Astrocladus dofleini* and the holotype of *Astrocladus pardalis* (ZSM 20000898) and the holotype of *Astrocladus annulatus* (UMUTZ-Echi-Oph-26) are preserved in ethanol, but their fixation methods are unknown.

The specimens for molecular analysis were collected by commercial fishing lines of at Minabe Port, Wakayama Prefecture (2 specimens), by fisherman with small trawl at Mogi, Nagasaki Prefecture (4 specimens), by scuba diving at Hashiraguri, Oki Island, Shimane Prefecture (1 specimen) and by scuba diving near Misaki Marine Biological Station, Kanagawa Prefecture (2 specimens). The specimens were fixed and preserved in 70–99% ethanol. Permission for all of these newly collected nine specimens were approved by Miura fishermen's association (Kanagawa Prefecture) Mogi fishermen's association (Nagasaki Prefecture), Kishu-Hidaka fishermen's association (Wakayama Prefecture), Wakayama-Minami fishermen's association (Wakayama Prefecture), and JF Shimane fishermen's association (Shimane Prefecture).

### Morphological observation and terminology

Ossicles from *Astrocladus coniferus* (NSMT E-13118), *A. dofleini* (NSMT E-13124) and *A. exiguus* (NSMT E-13126) were isolated by immersion in domestic bleach (approximately 5% sodium hypochlorite solution), washed in deionized water, air-dried, and mounted on SEM stubs using double-sided conductive tape. The ossicles were observed and photographed with a Jeol JSM 5200LV SEM at Misaki Marine Biological Station, The University of Tokyo. A part of photographs were focus-stacked using the software CombineZM v. 1.0.0 (https://www.softpedia.com/get/Multimedia/Graphic/Graphic-Editors/CombineZM.shtml). The size of external ossicles, represented by the length of the longest axis, it is referred to as "length" in this study.

The morphological terms used to describe euryalid brittle stars follow *Thuy & Stöhr (2011)*, *Stöhr, O'Hara & Thuy (2012)*, *Okanishi (2016)* and *Hendler (2018)*. The granular

**Table 1 Examined specimens of *Astrocladus* species including outgroup.**

| Specimen no. | Species | Catalogue. number | Locality | Depth (m) | Date of sampling | Type status | COI Access. no. | References |
|---|---|---|---|---|---|---|---|---|
| 1 | *Astrocladus annulatus* | UNUTZ-Ooh-26 | Off Misaki, Sagami Bay, Kanagawa | – | – | Holotype | – | *Matsumoto (1912a)* |
| 2 | *Astrocladus coniferus* | NSMT E-13118 | Sagami Bay, Hama Moroiso, Koajiro, Kanagawa, Japan | 0.5–1.0 | 2018/3/12 | Non type | LC546637 | This study |
| 3 | *Astrocladus coniferus* | NSMT E-13119 | Hashiraguri, Oki Island, Dohgo, Shimane, Japan | ca. 20 | 2010/7/15 | Non type | LC546640 | This study |
| 4 | *Astrocladus coniferus* | ZSM 20000897 | Kagoshima Bay, Kagoshima | ca. 30 | 1880/8 | Lectotype | – | *Döderlein (1902)* |
| 5 | *Astrocladus pardaris* | ZSM 20000898 | Sagami Bay, Kanagawa | – | – | Holotype | – | *Döderlein (1902)* |
| 6 | *Astrocladus dofleini* | NSMT E-13120 | Tachibana Bay, Off Mogi, Nagasaki, Japan | ca. 40 | 2019/2/7 | Non type | LC546641 | This study |
| 7 | *Astrocladus dofleini* | NSMT E-13121 | Tachibana Bay, Off Mogi, Nagasaki, Japan | ca. 40 | 2019/2/7 | Non type | LC546642 | This study |
| 8 | *Astrocladus dofleini* | NSMT E-13122 | Tachibana Bay, Off Mogi, Nagasaki, Japan | ca. 40 | 2019/2/7 | Non type | LC546643 | This study |
| 9 | *Astrocladus dofleini* | NSMT E-13123 | Tachibana Bay, Off Mogi, Nagasaki, Japan | ca. 40 | 2019/2/7 | Non type | LC546644 | This study |
| 10 | *Astrocladus dofleini* | NSMT E-5480 | Off Minabe, Wakayama | ca. 80 | 2006/3/10 | Non type | AB605105 | *Okanishi & Fujita (2013)* |
| 11 | *Astrocladus dofleini* | NSMT E-10749 | Off Kuji, Hitachi, Ibaraki | – | 2016/9/30 | Non type | LC546638 | This study |
| 12 | *Astrocladus dofleini* | NSMT E-13124 | Sagami Bay, Mouthe of Koajiro Bay, Kanagawa, Japan | 15 | 2018/6/6 | Non type | – | This study |
| 13 | *Astrocladus dofleini* | ZMB 5923 | Sagami Bay, Kanagawa | – | – | Paralectotype | – | *Döderlein (1910)*; *Doflein, 1906* |
| 14 | *Astrocladus dofleini* | ZSM 20000901/1 | Okinose, Sagami Bay, Kanagawa | 600 | 1904-1905 | Lectotype | – | *Döderlein (1910)*; *Doflein, 1906* |
| 15 | *Astrocladus dofleini* | ZSM 20000901/2 | Okinose, Sagami Bay, Kanagawa | 250 | 1904-1905 | Paralectotype | – | *Döderlein (1910)*; *Doflein, 1906* |
| 16 | *Astrocladus dofleini* | ZSM 20000901/3 | Okinose, Sagami Bay, Kanagawa, St. 5 | 600 | 1904 | Paralectotype | – | *Döderlein (1910)*; *Doflein, 1906* |
| 17 | *Astrocladus dofleini* | ZSM 20000901/4 | Okinose, Sagami Bay, Kanagawa | 600 | – | Paralectotype | – | *Döderlein (1910)*; *Doflein, 1906* |
| 18 | *Astrocladus exiguus* | NSMT E-6265 | Off Yaku-shima Island, Kagoshima | 155–170 | 2008/8/2 | Non type | AB605106 | *Okanishi & Fujita (2013)* |
| 19 | *Astrocladus exiguus* | NSMT E-13125 | Off Minabe, Wakayama | – | 2012/11/22 | Non type | LC546639 | This study |
| 20 | *Astrocladus exiguus* | NSMT E-13126 | Off Minabe, Wakayama | – | 2019/4/4 | Non type | LC546645 | This study |
| 21 | *Asteromoana muricatopatella* | NSMT E-5619B | Off Yaku-shima Isl., Kagoshima. | 140 | 2007/9/26 | Non type | AB605100 | *Okanishi & Fujita (2013)* |

| Specimen no. | Species | Catalogue. number | Locality | Depth (m) | Date of sampling | Type status | COI Access. no. | References |
|---|---|---|---|---|---|---|---|---|
| 22 | *Asteroporpa australiense* | MV F99691 | Wanganella Bank, New Zealand. | 254–259 | 2003/5/28 | Non type | AB605098 | *Okanishi & Fujita (2013)* |
| 23 | *Astroboa arctos* | NSMT E-6718 | Off Minabe, Wakayama | ca. 30 | 2009/2/26 | Non type | AB605101 | *Okanishi & Fujita (2013)* |
| 24 | *Astroboa nigrofurcata* | MNHN IE-2013-8003 | Northern Pacific | 143–173 | 1998/8/7 | Non type | AB758505 | *Okanishi & Fujita (2013)* |
| 25 | *Astrodendrum sagaminum* | NSMT E-5645 | Sagami Sea, Kanagawa | 681–716 | 2007/11/28 | Non type | AB605109 | *Okanishi & Fujita (2013)* |
| 26 | *Astrothorax misakiensis* | NSMT E-6266 | Off Toshima Isl., Tokyo | 266–312 | 2008/8/6 | Non type | AB605116 | *Okanishi & Fujita (2013)* |

**Note:**
COI accession numbers are lodged at the DNA Data Bank of Japan. See referees for the detailed information. Unknown data are shown by "–". Abbreviations: MNHN, Muséum national d'Histoire naturelle, Paris, France; MV, Museum Victoria, Melbourne, Australia; NSMT, the National Museum of Nature and Science, Tsukuba, Japan; UMUT, The University Museum, The University of Tokyo, Japan; ZMB, Museum für Naturkunde der Humboldt-Universität zu Berlin, Germany; ZSM, the Zoologische Staatssammlung München, Germany.

external ossicle is referred to as a "granule" in this article. Taxonomic arrangement follows *Christodoulou et al. (2019)*.

## DNA extraction, amplification and sequencing

Genomic DNA was extracted using DNeasy Blood and Tissue extraction kit (Qiagen, Hilden, Germany) according to the manufacturer's protocol. We sequenced mitochondrial COI gene for phylogenetic analysis. The method of DNA extraction and PCR parameters followed *Okanishi & Fujita (2013)*. A primer set of COI005 (5′-TTAGGTTAAHW AAACCAVYTKCCTTCAAAG-3′) and COI008 (5-CCDTANGMDATCATDGCRT ACATCATTCC-3′) (*Okanishi & Fujita, 2013*) was used for PCR of COI. The optimum cycling parameters for those COI primers consisted of an initial denaturation step of 94 °C/2 min followed by 41 cycles of 94 °C/30 s,48 °C/90 s and 72 °C/60 s with final extension step at 72 °C/10 min was followed by storage at 4°C. The PCR products were separated from excess primers and oligonucleotides using Exo-SAP-IT (GEHealthcare, Chicago, IL, USA) by following manufacturer's protocol. All samples were sequenced bidirectionally and sequence products were run on the 3730xI DNA Analyzer of Misaki Marine Biological Station. The accession numbers of the DNA Data Bank of Japan (DDBJ) are shown in Table 1.

## Phylogenetic analysis

We sequenced 2 specimens of *Astrocladus coniferus*, 5 specimens of *Astrocladus dofleini* and 2 specimens of *A. exiguus*, but sequence data from a single specimens of *A. dofleini* (referred as *A. coniferus* in *Okanishi & Fujita (2013, 2018)*) and *A. exiguus* obtained by *Okanishi & Fujita (2013, 2018)* were also used. *A. exiguus* was added to the analysis to compare genetic distances and to determine the taxonomic rank of each phylogenetic group within *Astrocladus*. For outgroups, we selected 6 species of the subfamily Gorgonocephalinae with the shortest genetic distance from *Astrocladus* to avoid long

branch attraction (*Bergsten, 2005*; *Okanishi & Fujita, 2018*). These species were also used in a previous molecular phylogeny (*Okanishi & Fujita, 2013*, *2018*).

All sequences were aligned using the Clustal W algorithm in MEGA X (*Thompson, Higgins & Gibson, 1994*; *Kumar et al., 2018*). All missing sequences were scored as gaps. Overall average of substitution rate was computed using MEGA X according to the Kimura 2-parameter model (*Kimura, 1980*) to compare the evolutionary rate of each gene. The K2P, HKY and TN93 with gamma distributions were selected as best-fit models of the first, second and third codons, respectively (*Kimura, 1980*; *Hasegawa, Kishino & Yano, 1985*; *Tamura & Nei, 1993*), by using the "Find best fit models" option of MEGA X. Seaview ver. 4.3.0 (*Gouy, Guindon & Gascuel, 2010*) were used in preparing the data matrices in PHYLIP format.

Figtree v1.4.3 (http://tree.bio.ed.ac.uk/software/figtree/) was used in exploring tree files, in preparing NEWICK format and exploring alternative tree topologies. The phylogenetic tree was constructed with RAxML ver. 8.1.20 (*Stamatakis, 2014*) for maximum likelihood analysis (ML) to obtain bootstrap support values and with MrBayes v. 3.1.2 (*Ronquist & Huelsenbeck, 2003*) to obtain Bayesian posterior probabilities (BPP). The Markov-Chain Monte-Carlo (MCMC) process was run with four chains for 3,000,000 generations, with trees being sampled every 100 generations. The first 7,500 trees were discarded as burn-in. Data sets were partitioned by each codon for the maximum likelihood analysis to allow for separate optimization per-site substitution rates. The best-supported likelihood tree was found by performing 1,000 replications.

K2P genetic distances were computed within each clade and between clades using MEGA X to compare the evolutionary rate of COI gene.

## SYSTEMATICS

Order Euryalida *Gray, 1840*
Family Gorgonocephalidae *Ljungman, 1867*
Genus *Astrocladus Verrill, 1899*
Type species: *Euryale verrucosum Lamarck, 1816* (=*Astrocladus exiguus* (*Lamarck, 1816*))

**Diagnosis**

Arms five, branching. Number of arm segments less than six before first branch. No rows of calcareous plates on edge of disc margin. One madreporite situated on oralmost portion of interradial lateral disc. Arm spines present before the first branch. Hooklets on dorsal arms with one secondary tooth. Disc covered by variously shaped external ossicles and/or large tubercles (*Döderlein, 1927*; *Baker, 1980*; *McKnight, 2000*).

**Remarks**

Our molecular and morphological results have confirmed that *Astrocladus pardalis* (*Döderlein, 1902*) is a junior synonym of *A. coniferus* (*Döderlein, 1902*) which can be separated from *A. dofleini* (*Döderlein, 1910*). Therefore, 11 species are now known in this genus (see list of "Included species" below); *A. annulatus*, *A. coniferus*, *A. dofleini* and *A. exiguus* are distributed in Japan.

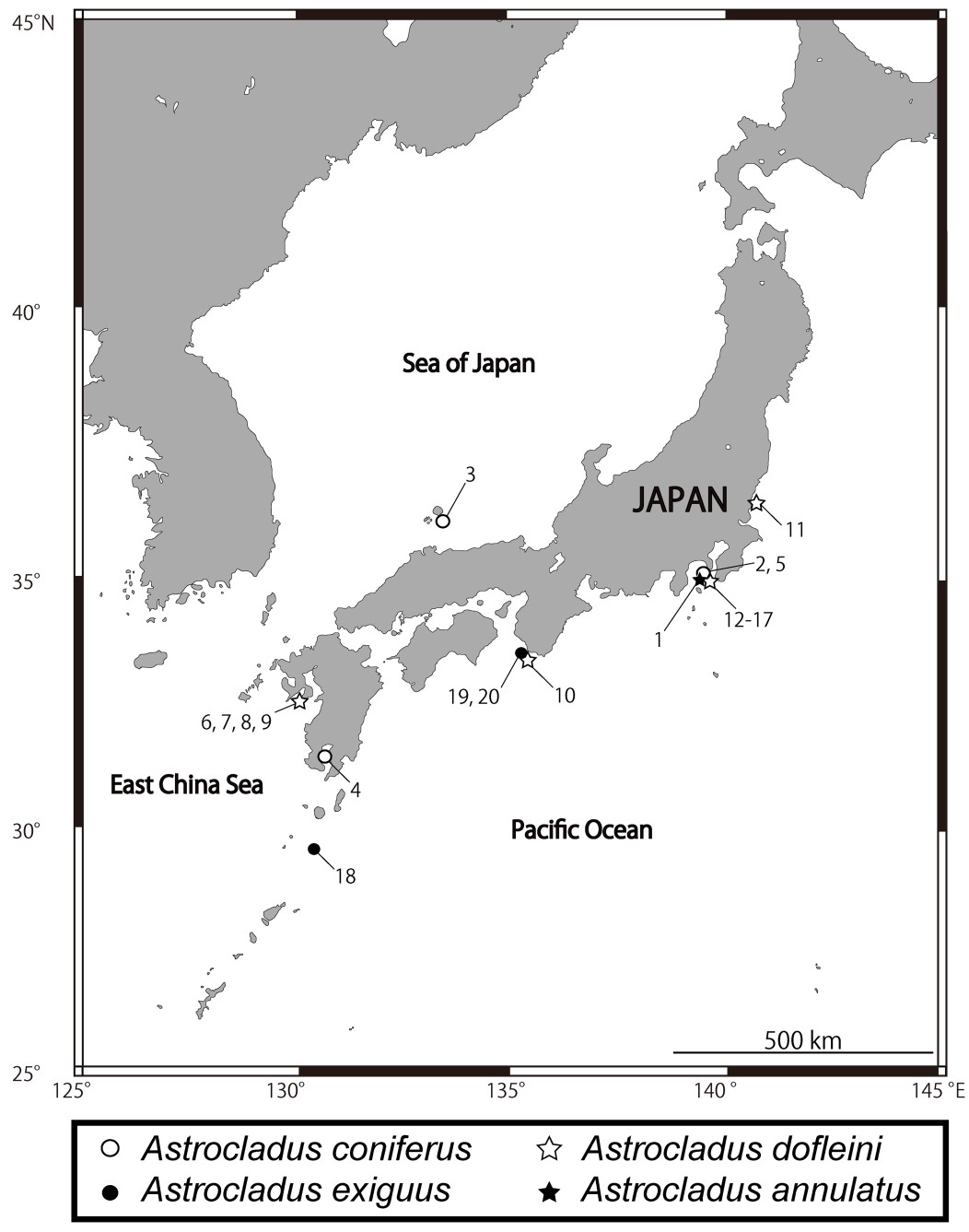

Figure 1 **Sampling sites of *Astrocladus annulatus*, *A. coniferus*, *A. dofleini* and *A. exiguus*.** Numerals indicate serial specimen number in Table 1.

### Included species

*A. africanus Mortensen, 1933*; *A. annulatus (Matsumoto, 1912a)*; *A. coniferus (Döderlein, 1902)*; *A. dofleini Döderlein, 1910*; *A. euryale (Retzius, 1783)*; *A. exiguus (Lamarck, 1816)*; *A. goodingi Baker, Okanishi & Pawson, 2018*; *A. hirtus, Mortensen, 1933*; *A. ludwigi (Döderlein, 1896)*; *A. socotrana Baker, Okanishi & Pawson, 2018*; *A. tonganus Döderlein, 1911*.

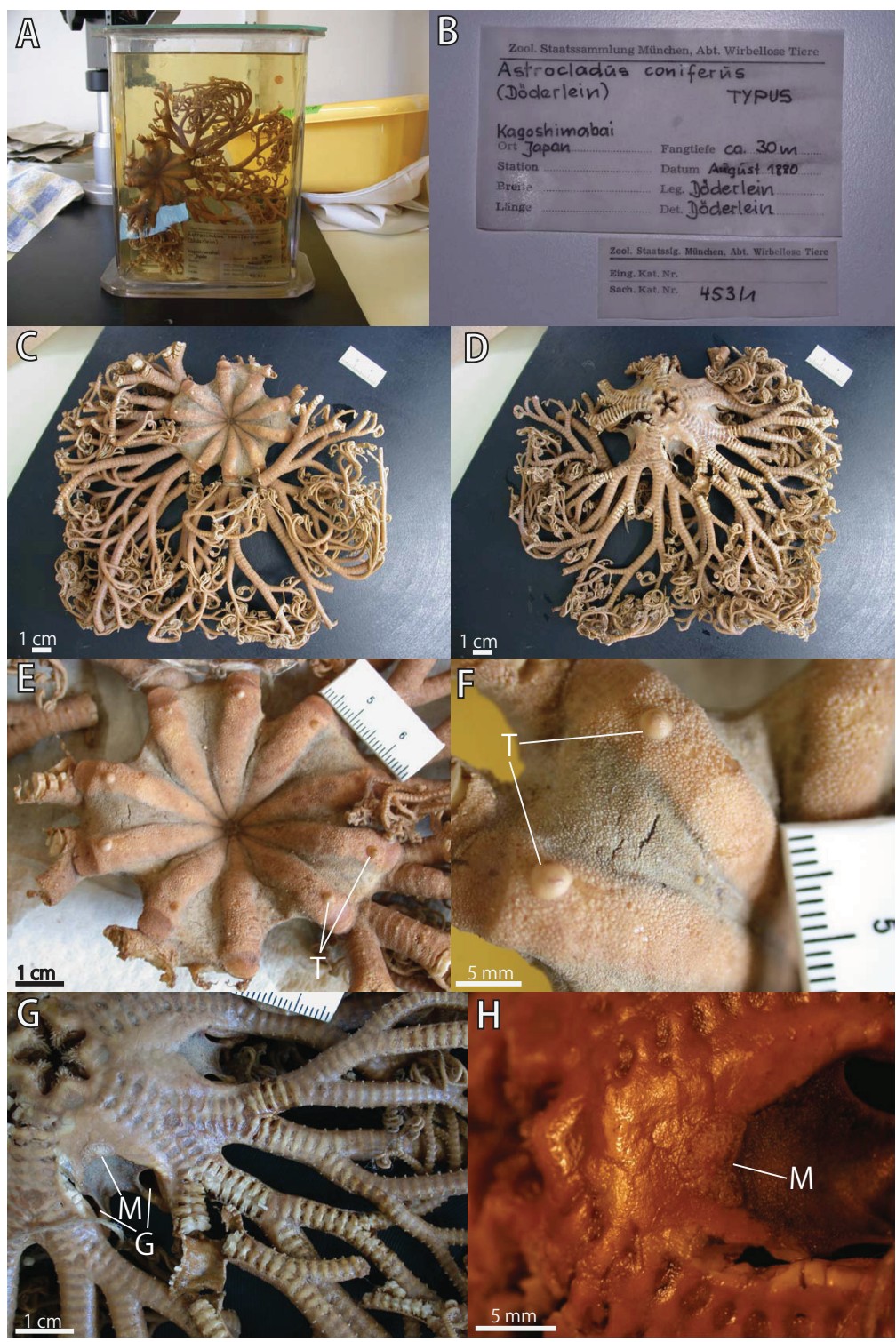

**Figure 2 *Astrocladus coniferus*, lectotype (ZSM 20000897).** (A) External view of lectotype bottle. (B) Original labels of the lectotype. (C) Dorsal view. (D) Ventral view. (E) Dorsal disc and proximal portion of arm. (F) Dorsal periphery of one radius of disc. (G) Ventral disc and proximal to middle portion of arm. (H) Ventral interradial disc. Abbreviations: G, genital slit; M, madreporite; T, large tubercle.

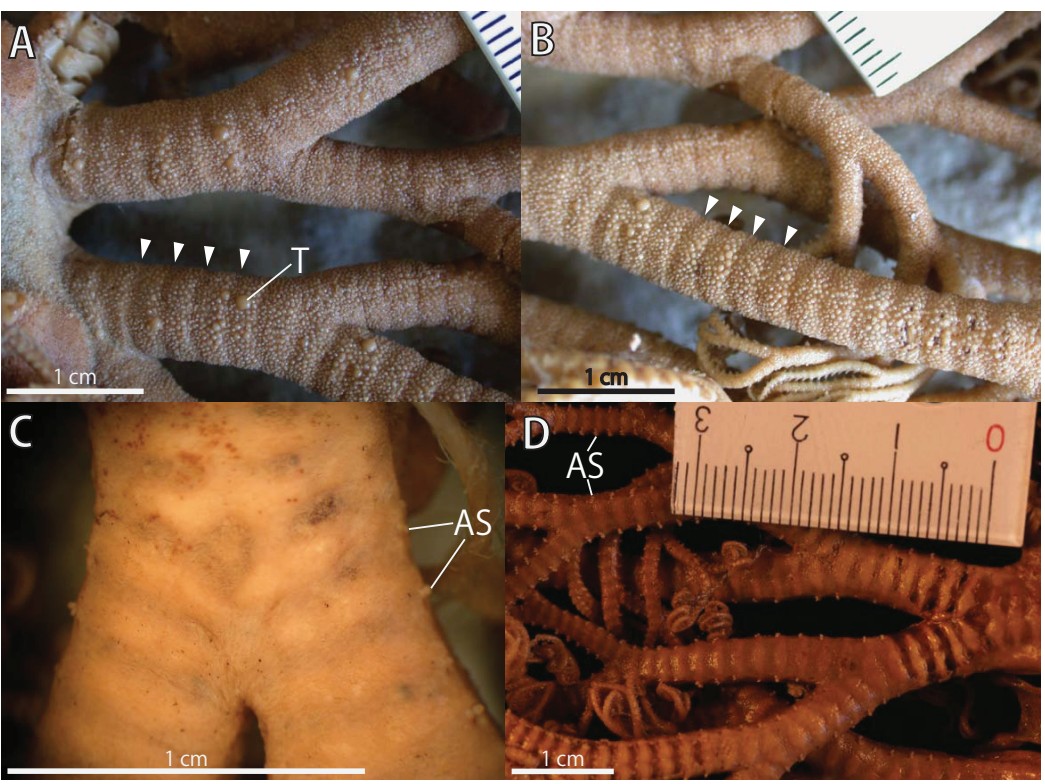

**Figure 3 *Astrocladus coniferus*, lectotype (ZSM 20000897).** (A) Dorsal proximal portion of arm. (B) Dorsal middle portion of arm. (C) Ventral proximal portion of arm. (D) Ventral middle portion of arm. Arrowheads indicate rows of hooklets on dorsal and lateral side of the arms (A and B). Abbreviations: AS, arm spine; T, large tubercle.

***Astrocladus coniferus*** (*Döderlein, 1902*)
(Figs. 1–7)

*Astrophyton coniferum* *Döderlein, 1902*, 325, 326; *Jangoux, De Ridder & Fechter, 1987*, 306.

*Astrocladus coniferus.*–Döderlein, 1911, 46–49, taf. 2, fig. 7, 7a; taf. 4, figs. 1–3a; taf. 7, 5, 6a, 16; *Döderlein, 1912*; Clark, H.L., 1915, 186; *Fedotov, 1926*, 473–477; *Murakami, 1944a*, 247–248; 1944b, 262; Djakonov, 1949, 50; 1954, 20; *Irimura, 1968*, 32; 1969, 39; 1981, 18–19; *Liao & Clark, 1995*, 170; *Ishida et al., 2001*, 8; *Fujita et al., 2004*, 192, 193; *Okanishi et al., 2011*, 378, 379, fig. 6G–J.

*Astrocladus coniferus coniferus.*–Irimura, 1982, 9–11, fig. 5, pl. 1(3); *Rho & Shin, 1987*, 209, 211; *Shin, 1992*, 118, 121; *Shin & Rho, 1996*, 389; *Ishida et al., 2001*, 8.

*Astrocladus coniferus pardalis.*–*Saba, Tomida & Kimoto, 1982*, 27. pl. 14 (2, 3); *Shin & Rho, 1996*, 390; *Ishida et al., 2001*, 8.

*Astrophyton cornutum.*–Clark, H.L., 1911: 293.

*Astrophyton pardalis* *Döderlein, 1902*, 323; Clark, H.L., 1911, 293–294; *Jangoux, De Ridder & Fechter, 1987*, 308.

*Astrocladus coniferus* var. *pardalis*—Matsumoto, 1917, 77; *Fedotov, 1926*, 473–477; *Djakonov, 1954*, 20.

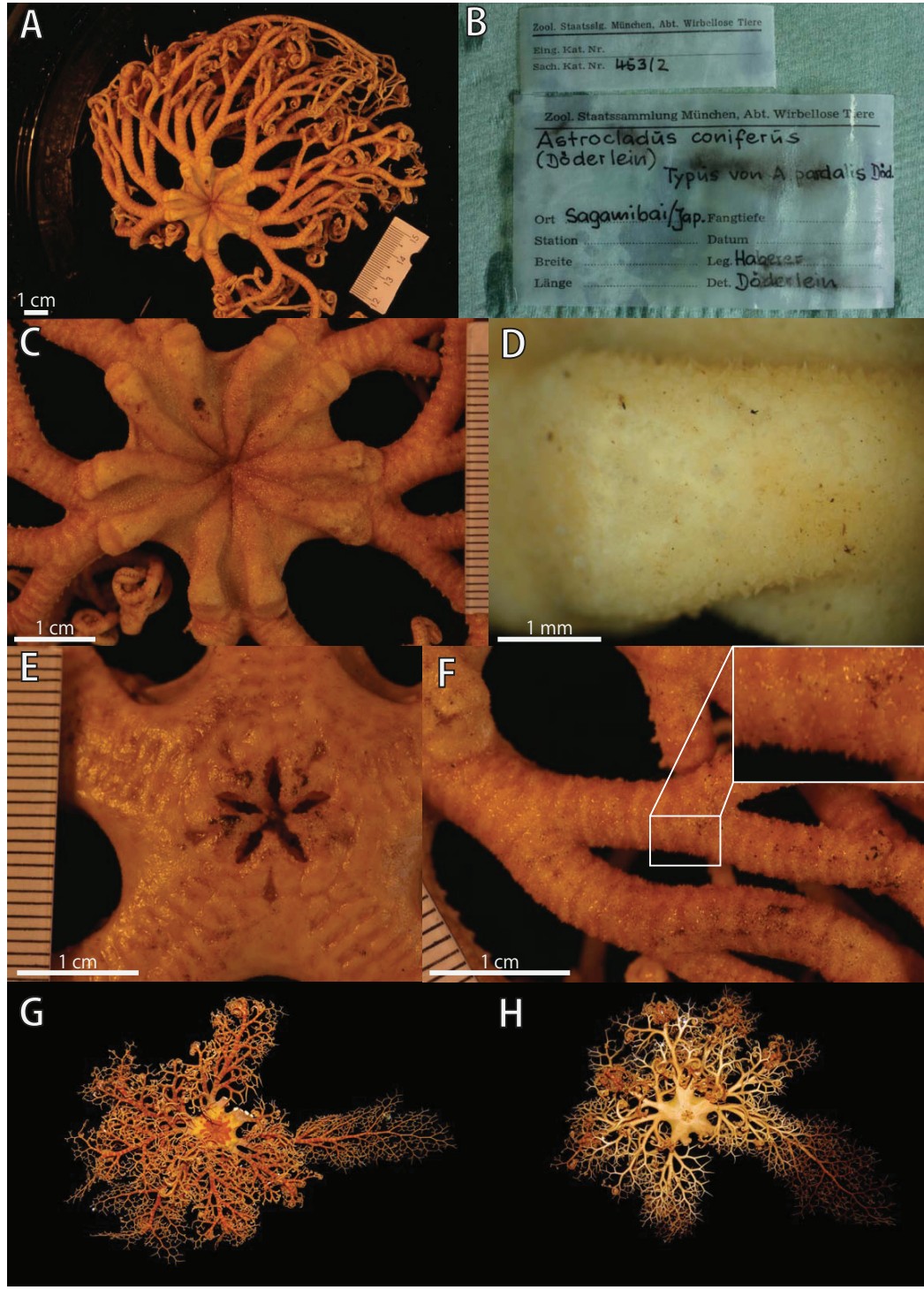

**Figure 4** *Astrocladus coniferus,* holotype of *Astrophyton pardalis* (ZSM 20000898) (A–F) and NSMT E-13118 (G and H). (A) Dorsal view. (B) Original labels of the holotype. (C) Dorsal disc and proximal portion of arm. (D) Dorsal periphery of radial shield (E). Ventral disc. (F) Dorsal proximal portion of arm, partly enlarged. (G and H) Live specimens, dorsal (G) and ventral (H) views.

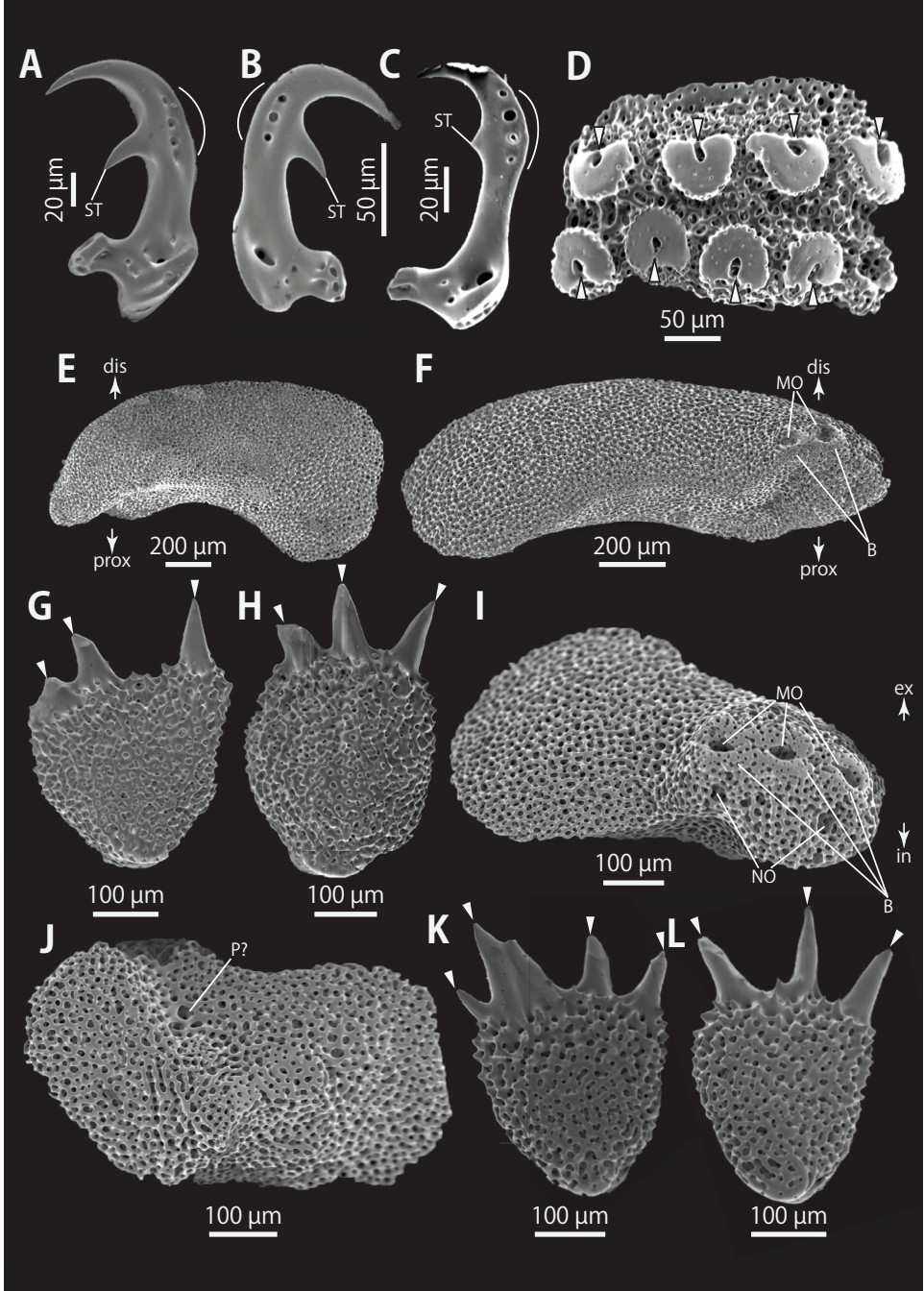

**Figure 5** *Astrocladus coniferus* (NSMT E-13118). SEM photographs of ossicles. (A–C) Hooklets on proximal (A), middle (B) and distal (C) portion of arm, arcs indicate reticular structure. (D) Hooklet-bearing plate on proximal portion of arm. (E and F) Lateral arm plates on proximal portion of arm, internal (E) and external (F) views. (G and H) Arm spines from proximal portion of arm, inner most (G) and second inner most (H). (I and J) Lateral arm plates on middle portion of arm, distal (I) and internal (J) views. (K and L) Arm spines on middle portion of arms, inner most (K) and second inner most (L). Arrowheads indicate articulations for hooklets (D) and terminal projections (G, H, K and L). Orientations: dis, distal side; ex, external side; in, internal side; prox, proximal side. Abbreviations: B, border structure; MO, muscle opening; NO, nerve opening; P, perforation; ST, secondary tooth.

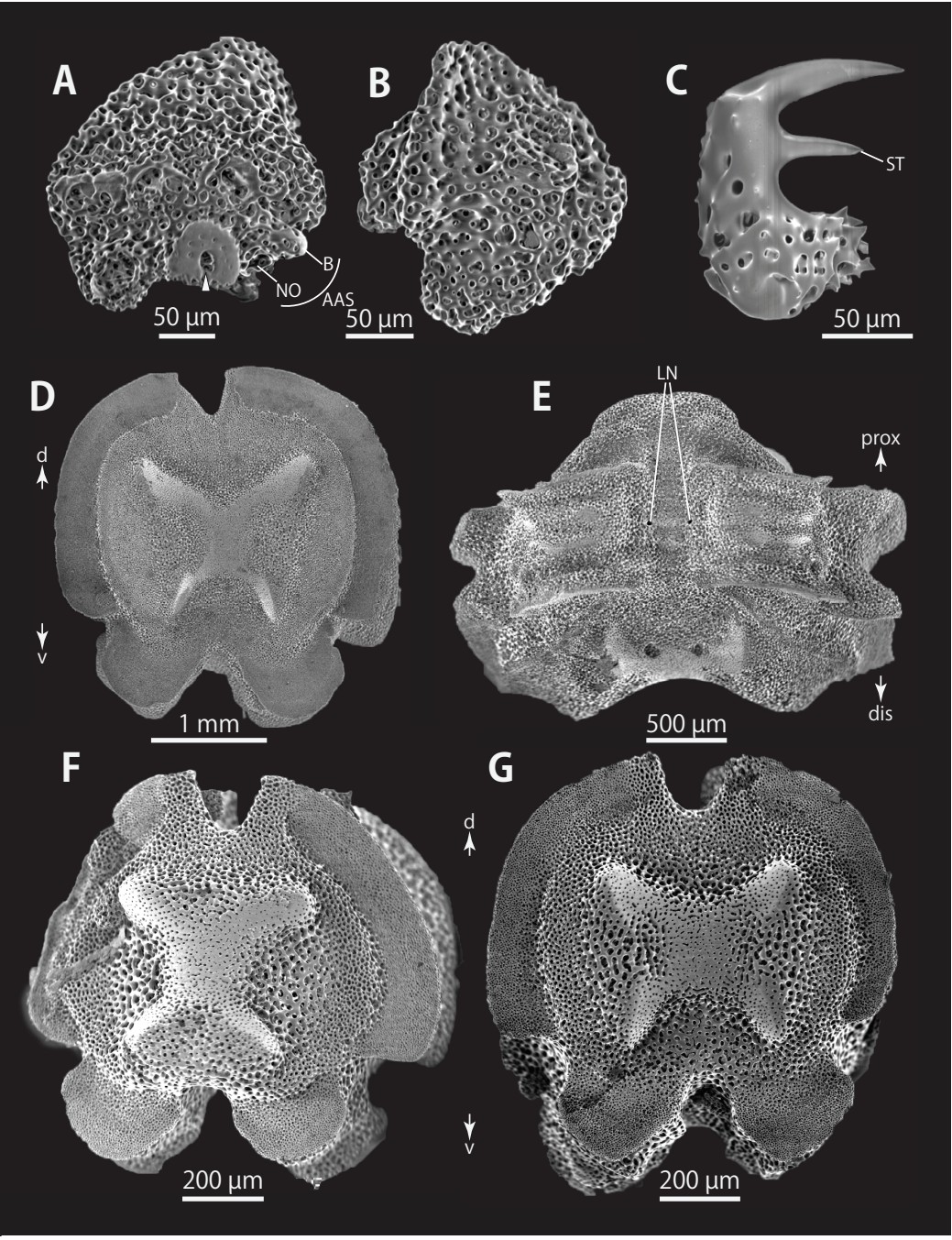

**Figure 6** *Astrocladus coniferus* (NSMT E-13118). SEM photographs of ossicles. (A and B) Lateral arm plates on distal portion of arm, external (A) and internal (B) views. (C) Hook-shaped arm spine on distal portion of arm. (D–G) Vertebrae from proximal (D and E) and middle (F and G) portion of arm, distal (D and G), ventral (E) and proximal (F) views. An arrowhead indicates articulation for hooklet (A). Orientations: d, dorsal side; dis, distal side; prox, proximal side; v, ventral side. Abbreviations: AAS, articulation for arm spine; B, border structure; LN, passage of lateral canal; NO, nerve opening; ST, secondary tooth.

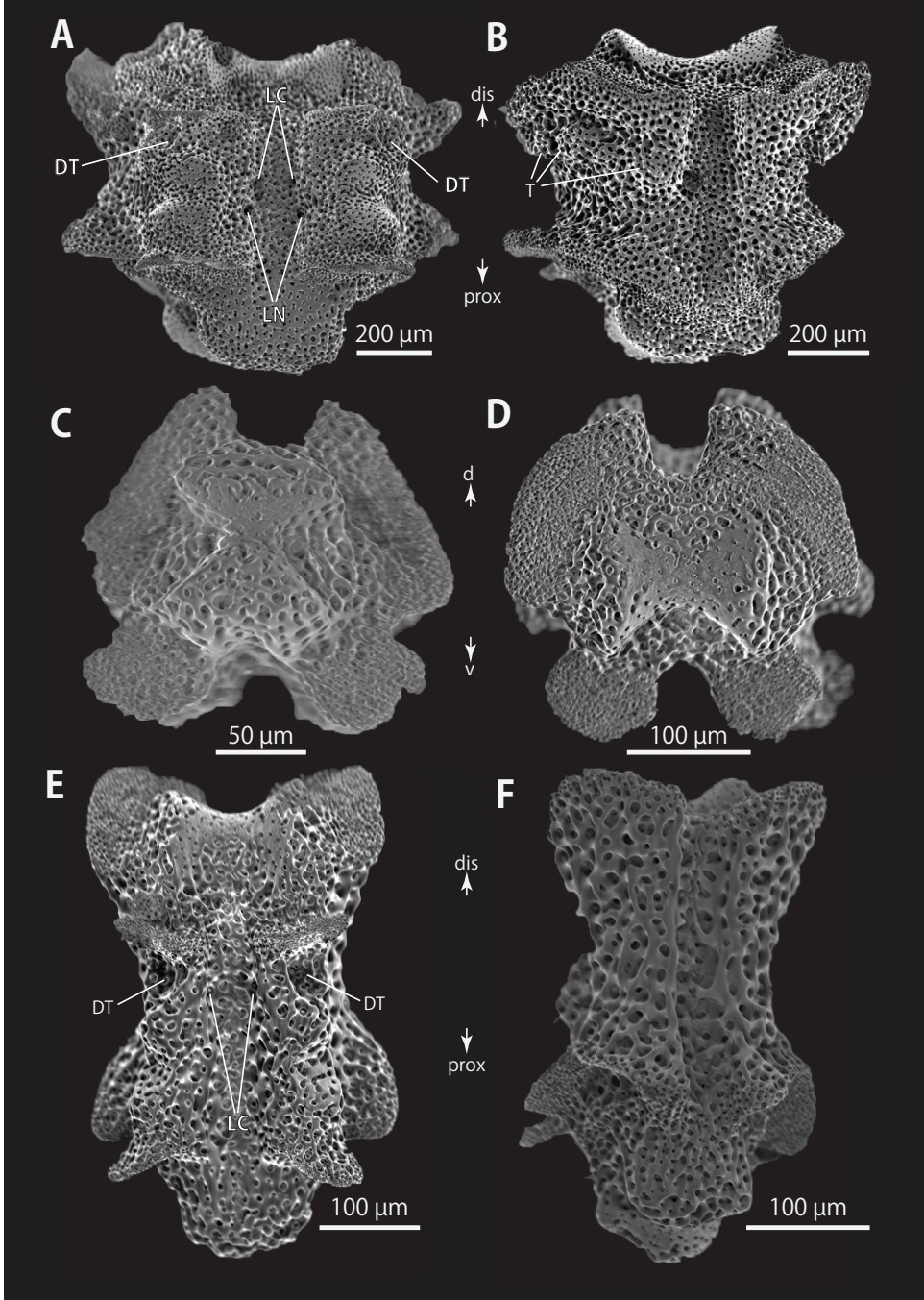

**Figure 7 *Astrocladus coniferus* (NSMT E-13118). SEM photographs of ossicles.** Vertebrae from middle (A and B) and distal (C–F) portion of arm, ventral (A and E), dorsal (B and F), proximal (C) and distal (D) views. Orientations: d, dorsal side; dis, distal side; prox, proximal side; v, ventral side. Abbreviations: DT, depression for tentacle; LC, passages of lateral canal; LN, passages of lateral nerve; T, tubercle.

(Non) *Astrocladus coniferus coniferus.*–*Saba, Tomida & Kimoto, 1982*, 26–27, pl. 14(1) (=*Astrocladus dofleini*)

(Non) *Astrocladus coniferus.*—*Matsumoto, 1917*, 77–79, fig. 23c; *Irimura, 1990*, 75, an unamed pl.; *Fujita & Kohtsuka, 2003*, 27, 28, pl. 1B (= *Astrocladus dofleini*)

(Non) *Astrocladus coniferus* var. *pardalis.*–*Irimura, 1982*, 12–13, fig. 7, pl. 4(4) (= *Astrocladus dofleini*)

**Notes on lectotype**
In the original description (*Döderlein, 1902*), *Astrophyton coniferum* (=*Astrocladus coniferus*) was thought to have been described based on two specimens which are listed in a table (*Döderlein, 1902*, P326). They were collected in Kagoshima Bay at, ca. 30 m depth and subsequently, one of them was figured by the same author in 1911 as "Typus" of *Astrocladus coniferus* (*Döderlein, 1911*, Taf 4, 2-2a). This is in accordance with §75.4 of the International Code of Zoological Nomenclature, and it can be regarded as the lectotype. The morphological traits of ZSM 20000897 concur with this figured specimen. Therefore, ZSM 20000897 is the lectotype.

**Type material examined**
ZSM 20000897, the lectotype of *Astrophyton coniferum* (*Döderlein, 1902*) (*Astrocladus coniferus*), Kagoshima Bay, ca 30 m, Japan, 1880 August (Figs. 2A and 2B). ZSM 20000898, the holotype of *Astrophyton pardalis* (*Döderlein, 1902*), collected by Karl Haberer, Sagami Bay, Japan, date unknown (Fig. 4B). In the original description (*Döderlein, 1902*), *Astrophyton pardalis* was implied to be described based on the single specimen, listed in a table (*Döderlein, 1902*, P326). Therefore, this specimen (ZSM 20000898) is the holotype by monotypy (ICZN Article 73.1.2.; see also *Jangoux, De Ridder & Fechter, 1987*)

**Other material examined**
NSMT E-13118, Sagami Bay, Hama Moroiso, Kanagawa, Japan, 0.5–1.0 m, 12 March 2018, collected by Hisnaori Kohtsuka, scuba. NSMT E-13119, Hashiraguri, Oki Island, Dohgo, Shimane, Japan, 20 m, 15 July 2010, collected by Hisanori Kohtsuka, scuba.

**External morphology of the lectotype (ZSM 20000897)**
*Disc.* Disc five-lobed with notched interradial edges, 60 mm in diameter (Fig. 2C). Dorsal disc wholly covered by external ossicles in contact with each other. Radial shields completely covered by granules and conical external ossicles, approximately 110–450 μm in length; other areas covered by smaller granules, approximately 100 μm in length (Fig. 2F). Radial shields tumid, bar-like, approximately 50 mm in length, width gradually decreasing from 10 mm at disc periphery to 2.5 mm almost at disc center (Fig. 2E). One large conical tubercle on peripheral edge of each radial shield, 2.5–3.4 mm in length (Fig. 2E).

Ventral surface of disc completely covered by skin and polygonal plate-like external ossicles, fully in contact, approximately 600 μm in length (Fig. 2G). Oral shields, adoral shields, oral plates and ventral arm plates completely concealed by ossicles (Fig. 2G). Teeth uniformly spiniform, situated on top of dental plates (Fig. 2G). Teeth arranged in a

cluster covering ventral-most part of dental plate approximately 10 in number (Fig. 2G), and in a vertical line, on dorsal part of dental plates, 3 or 4 in number. Spiniform oral papillae situated in 1 or 2 transverse rows on ventral edge of each oral plate, 4 to 5 in number (Fig. 2G). Size of teeth varying in position on jaw, approximately 400–1000 μm in length and oral papillae approximately 400 μm in length (Fig. 2G).

Interradial surface of lateral disc covered by granules fully in contact, approximately 100 μm in length (Figs. 2G and 2H). Two genital slits (5 mm long and 1–3 mm wide) in each interradius (Fig. 2G). One large, elliptical madreporite situated on ventral interradius, approximately 5.5 mm in width and 3.75 mm in length (Fig. 2H).

*Arms*. Arms branching. On proximal portion before first branch, arm 8.5 mm wide with arched dorsal surface and flattened ventral surface (Fig. 3A). Between first branch and second branch, arm width abruptly decreases to 5.8 mm. Subsequently, arms tapering gradually toward arm tip (Fig. 3).

On dorsal and lateral surface, each arm segment covered by single annular row of large hooklet-bearing plates (Figs. 3A and 3B). Before second branch, each plate separated by granules. Plates fully in contact from third branch and subsequent distal segments (Fig. 3B). With exception of hooklet-bearing plates dorsal and lateral surface of arm completely covered by granules, fully in contact, approximately 200–400 μm in length (Fig. 3A). Before first branch, ventral surface covered by polygonal plate-like external ossicles, similar to those on ventral disc, approximately 100 μm in length (Fig. 3C). After first branch, ossicles become into round granules, slightly in contact, and decreasing in size gradually toward arm tip (Fig. 3D). With exception of the articulations with arm spines and/or hooklets, lateral arm plates and ventral arm plates completely concealed by external ossicles on entire arm (Figs. 3C and 3D). Tentacle pore with single arm spine before first branch; 2 or 3 spines after first branch; up to 4 spines on subsequent pores (Figs. 3C and 3D). Number of arm spines decrease gradually to 2 towards arm tip. Arm spines approximately one-seventh to one-eighth (ca. 12–13%) of length of the corresponding arm segment on proximal portion of arm, and one-thirds to one-fourth length of the corresponding arm segment on middle to distal region of arm (Figs. 3C and 3D).

*Color*. Uniformly dull brown in ethanol preserved specimen (Figs. 2C and 2D).

**Description of other materials**

*External morphology of ZSM 20000898*: Disc approximately 30 mm in diameter (Figs. 4A and 4C). The external ossicles on radial shield conical with acute tip (Fig. 4D). No large tubercle on periphery of radial shields (Fig. 4C), tubercle present in lectotype of *Astrophyton coniferum* (ZSM 20000897). Teeth and oral papillae not spiniform, but granular (Fig. 4E).

*Living color of NSMT E-13118*: Dorsal disc (diameter = 30 mm) vivid orange with yellow patches, arms with yellow transverse bands on dorsal side (Fig. 4G). Ventral side of arms and disc uniformly creamy white, with orange arm tip (Fig. 4H).

*Ossicle morphology of NSMT E-13118*: All arm hooklets with single inner tooth and reticular structures (Figs. 5A–5C). Inner tooth on distal portion of arm smaller and more

**Table 2 Tabular key to the species of the *Astrocladus* in Japanese waters.** Tabular morphological characteristic key to *Astrocladus annulatus*, *A. coniferus*, *A. dofleini* and *A. exiguus* in Japanese waters.

| | Shape of external ossicles on dorsal disc | Shape of tubercles on dorsal disc | Position of tubercles on dorsal disc | Hooklet-bearing plates on proximal arm |
|---|---|---|---|---|
| *A. annulatus Matsumoto, 1912a* | Granule-shaped | Domed | On radial shield | Continuous |
| *A. coniferus (Döderlein, 1902)* | Conical on the peripheral part Granule-shaped on other parts | Conical | At periphery edge of radial shield | Discontinuous |
| *A. dofleini Döderlein, 1910* | Granule-shaped | Domed | Scattered on whole dorsal disc | Absent |
| *A. exiguus Lamarck, 1816* | Conical with acute thorny tip | Conical | Scattered on whole dorsal disc | Discontinuous |

rudimentary than those from proximal to middle portion of arm. Hooklet-bearing plates with 9 or 10 tubercle-shaped articulations for hooklets in proximal portion of arm; articulations forming 2 parallel rows (Fig. 5D). On proximal to middle portion of arm, lateral arm plates longer than high, curved to distal side (Figs. 5E, 5F, 5I and 5J). On proximal portion of arm, simple muscle openings besides border structures on distal edge (Fig. 5F), and on middle portion of arm, nerve openings on internal side of the muscle openings (Fig. 5I). One perforation present on internal side (Fig. 5J). On distal portion of arms, plate square, at least one nerve opening of articulation for arm spine beside border structure and one articulation for hooklet on distal side (Fig. 6A). No perforations recognizable on internal side (Fig. 6B). On proximal and middle portion of arm, arm spines ovoid, with three or four terminal projections, approximately one-third to one-fourth length of the height of arm spine (Figs. 5G, 5H, 5K, and 5L). In distal portion, arm spines transformed into hooks with one inner secondary tooth (Fig. 6C). Hook-shaped arm spines distinguished from hooklets by lack of reticular structure (Figs. 5A–5C and 6C).

All vertebrae with hourglass-shaped streptospondylous articulations (Figs. 6D, 6F, 6G, 7C, and 7D). In middle portion of arm, surfaces of lateral furrows smooth, with tubercle shaped ornamentations on dorsal side (Fig. 7B). In distal portion of arm, depressions for tube feet openings in distal part of ventral-lateral side of vertebrae but these are unrecognizable on proximal to middle portion of arm (Figs. 7E). A pair of channels for passage of lateral nerves opening inside ventral furrows (Figs. 6E, 7A, and 7E). In middle portion of arm channels for passage of lateral canals also opening on distal side of canals of lateral nerves (Fig. 7A). Channels for lateral nerve obscured in distal portion of arm (Fig. 7E).

**Distribution**

Many records of *Astrocladus coniferus* have been confused with those of *A. dofleini*. Therefore, only those that can be identified as *A. coniferus* by their figures and/or photographs are listed here. JAPAN: Kagoshima Bay, southwestern Japan, ca. 20 m (*Döderlein, 1902*, *1911*; type locality, Fig. 1); Sagami Bay, Kanagawa, central-eastern Japan, 1.5–130 m (*Döderlein, 1902*, *1911*; *Irimura, 1982*; This study, Fig. 1); Off Hachijojima Island and Ogasawara Islands, south-eastern Japan, 120–500 m (*Okanishi, O'Hara &*

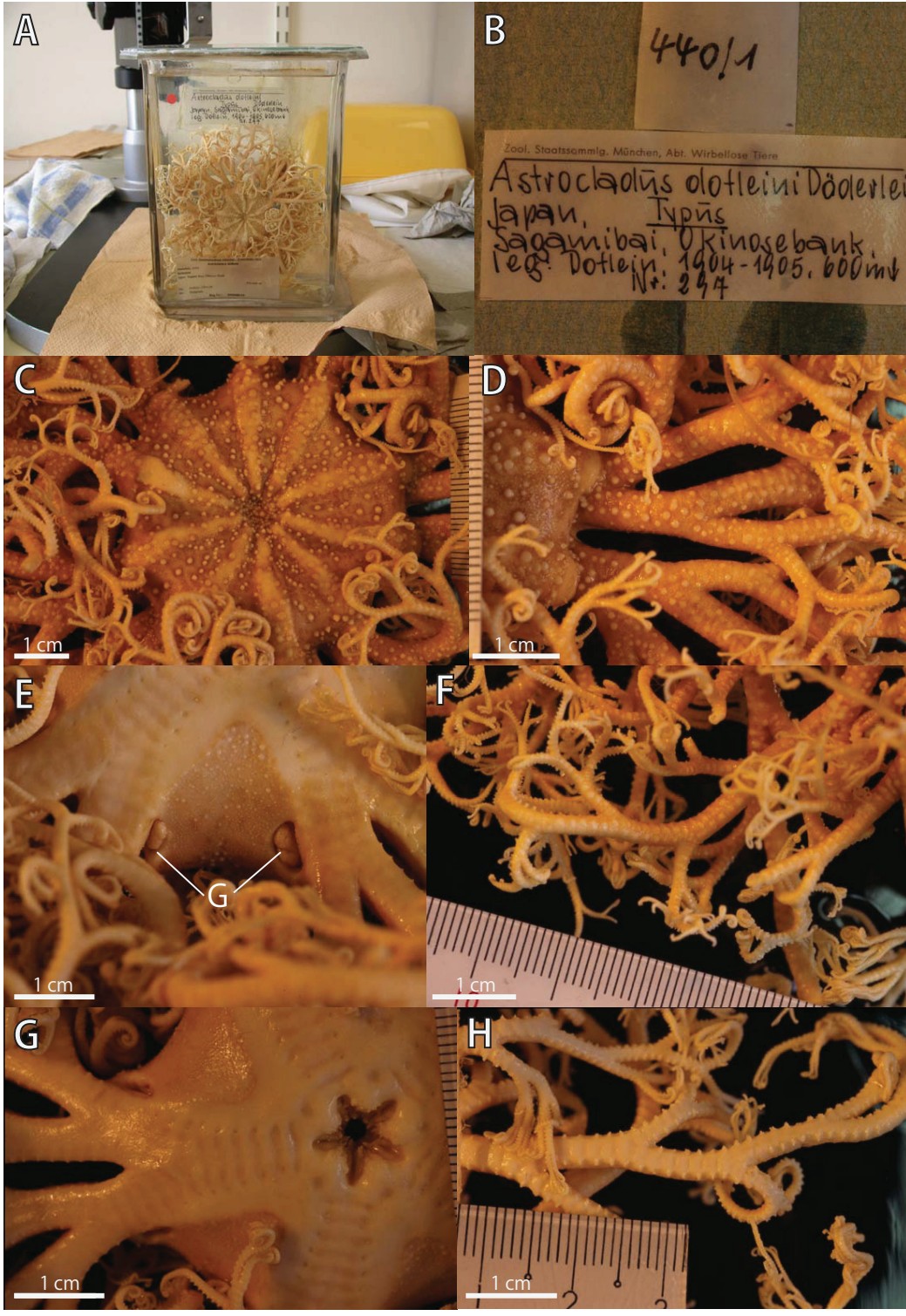

**Figure 8** *Astrocladus dofleini*, **lectotype (ZSM 20000901/1).** (A) External view of lectotype bottle. (B) Original labels of the lectotype. (C) Dorsal disc and proximal portion of arm. (D) Dorsal proximal portion of arm, partly enlarged in upper right. (E) Interradial ventral disc. (F) Dorsal middle to distal portion of arm. (G) Ventral disc and proximal portion of arm. (H) Ventral middle to distal tips of arm. Abbreviation: G, genital slit.

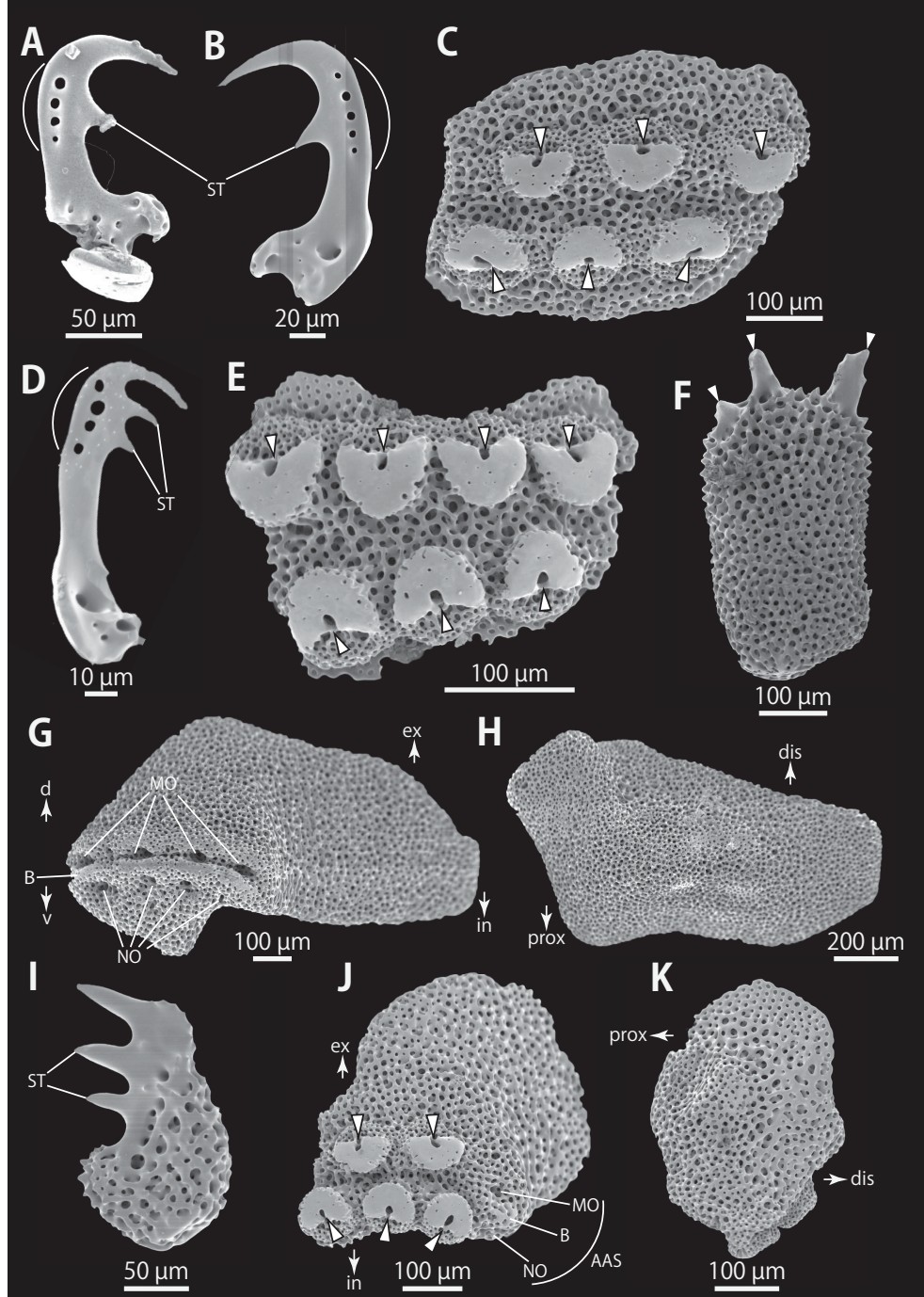

**Figure 9** *Astrocladus dofleini* (NSMT E-13124). SEM photographs of ossicles. (A, B and D) Hooklets on proximal (A), middle (B) and distal (D) portion of arms, arcs indicate reticular structure. (C and E) Hooket-bearing plate on proximal (C) and distal (E) portion of arm. (F and I) Arm spines on proximal (F) and middle (I) portion of arms. (G, H, J and K) Lateral arm plates on proximal (G and H) and middle (J and K) portion of the arms, external (G and J) and internal (H and K) views. Arrowheads indicate articulations of hooklets (C, E and J) and terminal projections (F). Orientations: d, dorsal side; dis, distal side; ex, external side; in, internal side prox, proximal side; v, ventral side. Abbreviations: AAS, articulation for arm spine; B, border structure; MO, muscle opening; NO, nerve opening; ST, secondary tooth.

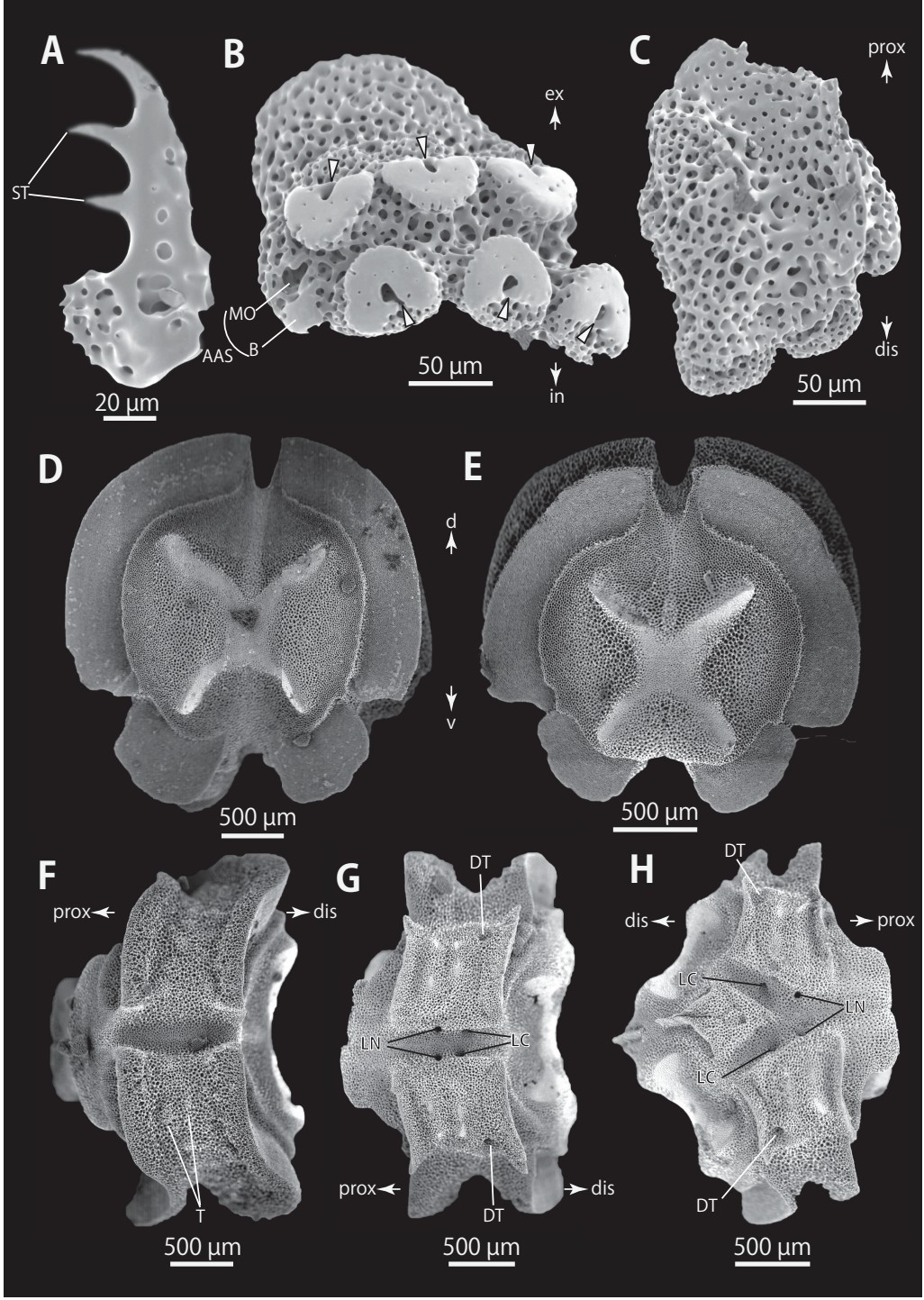

**Figure 10** *Astrocladus dofleini* (NSMT E-13124). **SEM photographs of ossicles.** (A) An arm spine from distal portion of arm. (B, C) Lateral arm plates on distal portion of arm, external (B) and internal (C) views. (D–H) Vertebrae from proximal portion of arm (H is branching vertebra), distal (D), proximal (E), dorsal (F) and ventral (G and H) views. Orientations: d, dorsal side; dis, distal side; prox, proximal side; v, ventral side. Arrowheads indicate articulations for hooklets. Abbreviations: B, boarder structure; DT, depression for tentacle; LC, passages of lateral canal; LN, passages of lateral nerve; MO, muscle opening; ST, secondary tooth; T, tubercle.

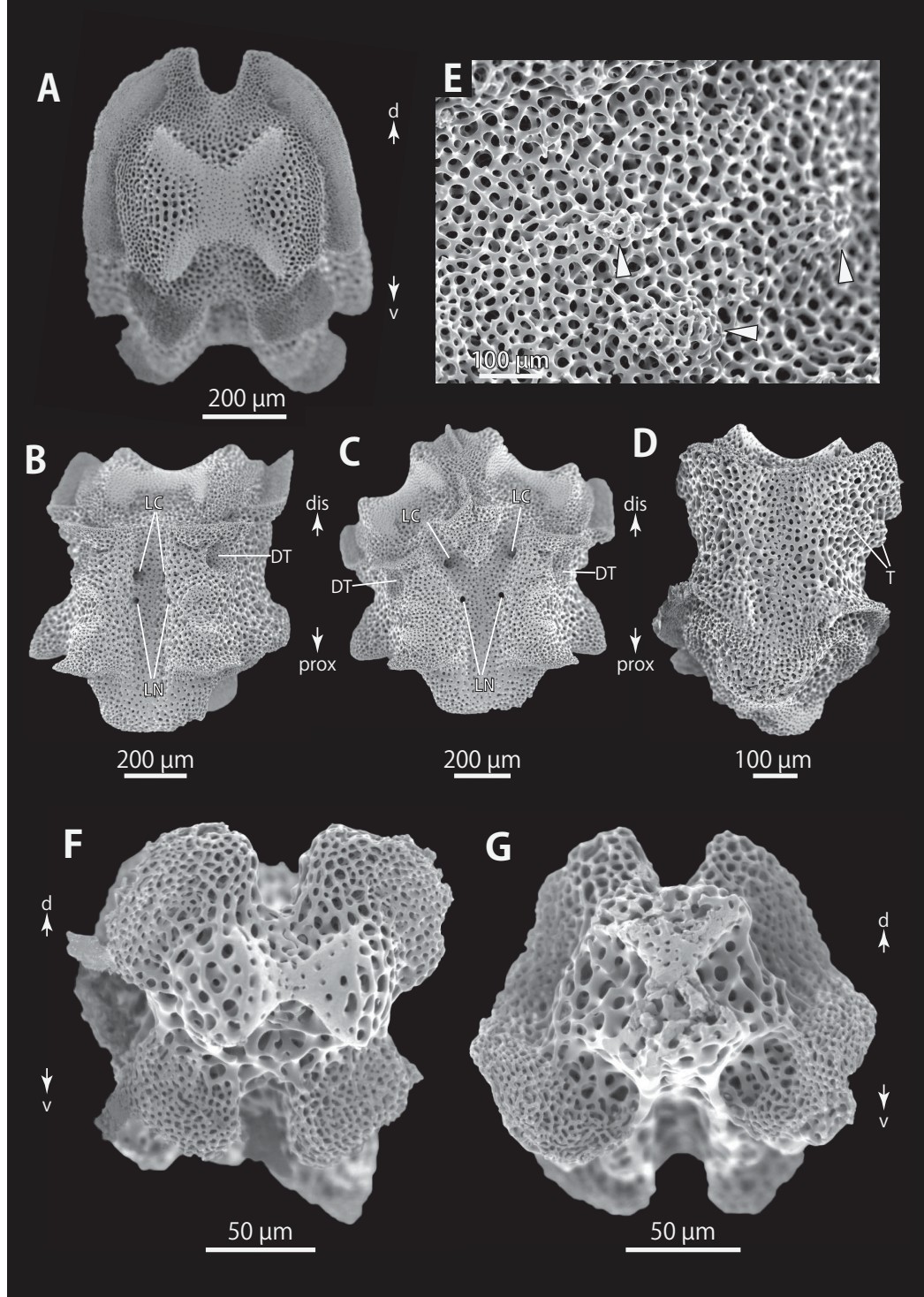

**Figure 11** *Astrocladus dofleini* (NSMT E-13124). **SEM photographs of ossicles.** (A–E) Vertebrae from middle portion of arm (C is branching vertebra), distal (A), ventral (B and C), dorsal (D) views, a part of (D) enlarged in (E). Vertebrae from distal portion of arm (F and G), distal (F) and proximal (G) views. Orientations: d, dorsal side; dis, distal side; prox, proximal side; v, ventral side. Arrowheads indicate tubercles on lateral furrow of vertebra. Abbreviations: DT, depression for tentacle; LC, passages of lateral canal; LN, passages of lateral nerve; T, tubercle.

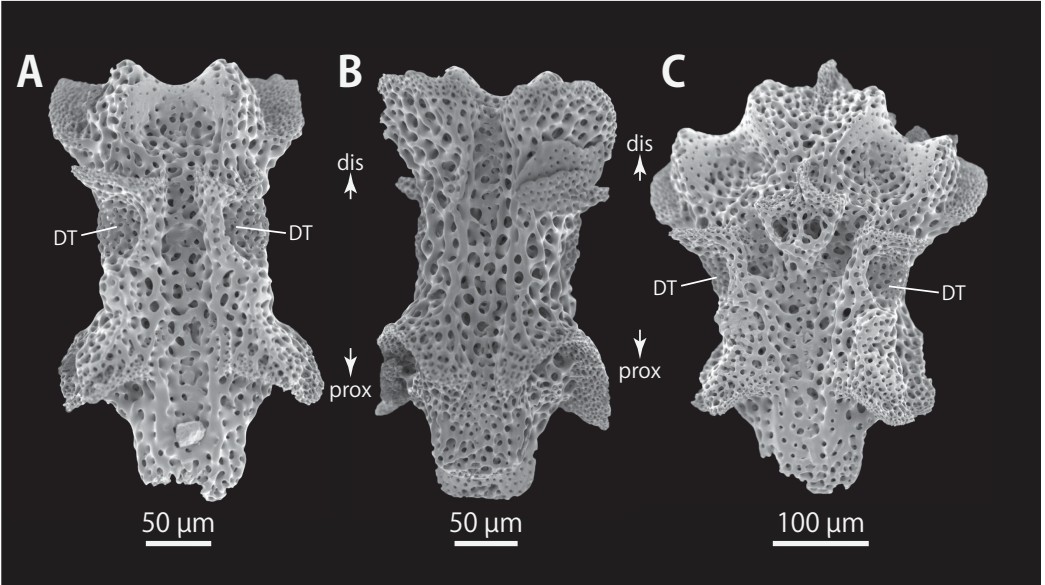

**Figure 12** *Astrocladus dofleini* **(NSMT E-13124).** SEM photographs of vertebrae from distal portion of arm (C is branching vertebra), ventral (A and C) and dorsal (B) views. Orientations: dis, distal side; prox, proximal side. Abbreviation: DT, depression for tentacle.

*Fujita, 2011*); Off Kii Nagashima, Mie, central Japan, depth unknown (*Saba, Tomida & Kimoto, 1982*); Hashiraguri, Oki Island, Shimane, western Japan, ca. 20 m (This study).

**Discussion**

In this study, we propose that *A. pardalis* is a junior subjective synonym of *A. coniferus* (see Remarks of *A. dofleini* for the details). *A. coniferus* can be distinguished from other congeners in having; granules and conical external ossicles on dorsal disc, and 0 or 1 conical large tubercles on the peripheral edge of each radial shield (Table 2).

*Astrocladus dofleini* Doderlein, 1910
(Figs. 8–12)

*Astrocladus dofleini Döderlein, 1910*, 256; 1911, 41–46, 106, fig. 9, taf. 2, fig. 6, taf. 3 figs. 1–4, taf. 4, figs. 4, 5, taf. 7, 15, 15b; 1927, 35, 94, taf. 3, fig. 2; *Guille, 1981*, 416, 417; *Jangoux, De Ridder & Fechter, 1987*, 306.

*Astrocladus coniferus dofleini.*—*Rho & Shin, 1987*, 209, 211; *Yi & Irimura, 1987*, 122, 123, fig. 2; *Shin, 1992*, 253; 1995, 117, 121; *Shin & Rho, 1996*, 391; *Ishida et al., 2001*, 8.

*Astrocladus coniferus var. dofleini.*—*Matsumoto, 1917*, 77–79, fig. 23; *Fedotov, 1926*, 473–477; Djakonov, 1949, 50; 1954, 20; *Irimura, 1968*, 32; 1979, 2; 1981, 19; 1982: 11, 12, fig. 6, pl. 4, figs. 5, 6.

*Astrocladus coniferus.*—*Matsumoto, 1917*, 77–79, fig. 23; *Irimura, 1990*, 75, pl; *Fujita & Kohtsuka, 2003*, 27, pl. 1B; *Kohtsuka et al., 2017*, 229–233, figs. 2–5 (Non *Astrocladus coniferus*).

*Astrocladus coniferus coniferus.*—*Saba, Tomida & Kimoto, 1982*, 26–27, pl. 14(1) (Non *Astrocladus coniferus*)

*Astrocladus coniferus* var. *pardalis.—Irimura, 1982*: 12, 13, text-fig. 7, pl. 4, fig. 4 (Non *Astrocladus pardalis*).

*Astrocladus verrucosus.–Verrill, 1899*, 369 (Non *Astrocladus verrucosus* (*Lamarck, 1816*)).

(Non) *Astrocladus dofleini.—Bomford, 1913*, 220, 221, pl. 13, fig. 1 (=*Astrocladus exiguus*).

## Notes on lectotype

In the original description (*Döderlein, 1910*), this species was based on a specimen in the Peabody Museum of Natural History (Yale University) which was reported as *A. verrucosus* by *Verrill (1899)*, plus several specimens collected from Japan. Subsequently, one of them was figured by *Döderlein (1911)*, and named the "Typus" of *Astrocladus coniferus*. In accordance with §75.4 of the International Code of Zoological Nomenclature, the author "unambiguously selected a particular syntype to act as the unique name-bearing type of the taxon", namely the lectotype. The morphological traits of one of the five syntypes, ZSM 20000901/1 concur with the specimen figured by *Döderlein (1911)*. Therefore, the specimen (ZEM 20000901/1) is designated as the lectotype, and the other four syntypes (ZSM 20000901/2, ZSM 20000901/3, ZSM 20000901/4 and ZMB 5923) as paralectotypes.

## Type material

ZSM 20000901/1, the lectotype of *Astrocladus dofleini Döderlein, 1902*, Okinose, Sagami Bay, ca 600 m, Japan, 1904–1905, collected by Franz Doflein (Fig. 8B). ZSM 20000901/2, a paralectotype of *Astrocladus dofleini Döderlein, 1910*, Okinose, Sagami Bay, 250 m, Japan, 1904–1905, collected by Franz Doflein, ZSM 20000901/3, a paralectotype of *Astrocladus dofleini Döderlein, 1910*, Okinose, Sagami Bay, Stat. 5, 600 m, Japan, 1904, collected by Franz Doflein, ZSM 20000901/4, a paralectotype of *Astrocladus dofleini Döderlein, 1910*, Okinose, Sagami Bay, 600 m, Japan. These four specimens of ZSM were probably collected with *Zuso-Maru* (see also *Doflein, 1906*). ZMB 5923, a paralectotype of *Astrocladus coniferus Döderlein, 1902*, Sagami Bay, depth unknown, Japan, collected by Karl Haberer.

## Other material examined

NSMT E-13124, 1 specimen, Sagami Bay, Mouth of Koajiro Bay, Kanagawa, Japan, ca. 15 m, 6 June 2018, collected by Hisanori Kohtsuka, Scuba. NSMT E-13120, NSMT E-13121, NSMT E-13122, NSMT E-13123, 4 specimens, Tachibana Bay, off Mogi, Nagasaki, Japan, ca. 40 m, 7 February 2018, collected by Hatsuyuki Takeshita, Small trawl. NSMT E-5480, 1 specimen, off Minabe, Wakayama, Japan, ca. 80 m, 10 March 2006, collected by Hajime Watabe, gill net. NSMT E-10749, 1 specimen, Kuji Port, Hitachi, Ibaraki, Japan, 36 30.50N, 140.38.40.E, depth unknown, 30 September, 2016, collected by fishery boat *Daisan-shouei-Maru*, fishing net.

## Description of external morphology of the lectotype (ZSM 20000901/1)

*Disc*. Disc circular with slightly notched interradial edges, 53 mm in diameter (Fig. 8C). Radial shields tumid (Fig. 8C). Dorsal disc wholly covered by granules in contact

each other and domed large tubercles (Fig. 8C). Radial shields covered by granules, approximately 200–330 µm in length (Fig. 8C). and large domed tubercles, 20–25 on each radial shield in number, each approximately 2.5–3.4 mm in length (Fig. 8C). Radial shields bar-like, approximately 25 mm in length, and the width gradually decreasing from 4.6 mm at disc periphery to 1.5 mm almost reaching to disc center (Fig. 8C).

Ventral surface of disc covered by polygonal plate-shaped external ossicles, fully in contact. Oral shields, adoral shields, oral plates and ventral arm plates concealed by ossicles (Fig. 8E). Teeth uniformly small, granule-like, situated on the top of dental plates, forming a cluster, approximately 6 to 8 in number (Fig. 8E). Oral papillae on the ventral edges of oral plates, forming a transverse row on each plate, 1 or 2 in number (Fig. 8E). Interradial surface of lateral disc covered by granules fully in contact, approximately 170 µm in length, and domed tubercles, approximately 200–500 µm in length (Fig. 8E). Two pore-like genital slits (6.5 mm long, 3 mm wide) in each interradius (Fig. 8E).

*Arms.* Arms branching. On proximal portion before first branch, arm 16 mm wide with an arched dorsal surface and flattened ventral surface (Figs. 8D and 8E). Between first branch and second branch, arm width abruptly decreasing to 10 mm. Subsequently, arms tapering gradually toward arm tip (Figs. 8F and 8H).

On dorsal and lateral surface, each arm segment covered by single annular row of hooklet-bearing plates. Before third branch, each plate separated by granules. The plates fully in contact from fourth branch and subsequent distal segments. With exception of hooklet-bearing plates, dorsal and lateral surface of arm completely covered by granules, fully in contact, approximately 100 µm in length. Before the first branch, ventral surface covered by polygonal and plate-shaped external ossicles, similar to those on ventral disc. After the first branch, the ossicles transforming granules, slightly in contact, and decreasing in size gradually toward the distal arm tip. With exception of the articulations with arm spines and/or hooklets, lateral arm plates and ventral arm plates concealed by external ossicles on entire arm (Figs. 8E, 8G, and 8H). Tentacle pore with single arm spine after the first branch; 1 or 2 spines after second branch; and up to 4 spines for the subsequent pores (Figs. 8G and 8H). Distally, the number of arm spines decreasing gradually to 2 toward arm tip. Arm spines approximately one-fourth to one-fifth of length (ca. 20–25%) of corresponding arm segment on proximal portion of arm, and one-thirds to one-fourth length of corresponding arm segment on middle to distal arm segment (Figs. 8G and 8H).

*Color.* Uniformly dull brown with whitish large tubercles in ethanol preserved specimen (Fig. 8).

### Description of other materials

*Ossicle morphology of NSMT E-13124*: Each hooklet on proximal and middle portion of arm with single inner secondary tooth, distal portion of arm with two inner secondary teeth. All hooklets with reticular structures (Figs. 9A, 9B, 9D). Hooklet-bearing plates with 6, 8 and 6 tubercle-shaped articulations for hooklets in proximal, middle and distal

portion of the arm, respectively; articulations forming 2 parallel rows (Figs. 9C, 9E, and 10B).

On proximal portion of arm, lateral arm plates long, with straight both proximal and distal edges (Figs. 9G and 9H); edges ellipse-shape in middle (Figs. 9J and 9K) and distal (Figs. 10B and 10C) portion of arms. No perforation observed on ventral side (Figs. 9H, 9K, and 10C). Pairs of simple nerve and muscle openings of articulation for arm spine with border structures on external edge (Figs. 9G, 9J, and 10B). On middle to distal portion of arms, lateral arm plates carrying 5 or 6 articulations for hooklets on external edge beside the articulation for arm spine (Figs. 9J and 10B). On proximal to middle portion of arm, arm spines ovoid, with three projections, approximately one-thirds to one-fifth to one-sixth length of the height of spine (Fig. 9F). In distal portion, arm spines transformed into hooks with two inner secondary teeth (Fig. 10A). Hook-shaped arm spines distinguished from hooklets on dorsal and lateral surface of arm by lack of reticular structure (Figs. 9A, 9B, 9D, and 10A).

All vertebrae with hourglass-shaped streptospondylous articulations (Figs. 10D, 10E, 11A, 12F, and 12G), and distal side of branching vertebra slightly wider than non-branching vertebra due to their possession of 2 articulation surfaces (Figs. 10H, 11C, and 12C). Lateral furrows of vertebrae ornamented by tubercles in proximal to middle portion of arm, but smooth in distal portion of arm (Figs. 10F, 11D, and 12B). Depressions for tube feet openings in distal part of ventral-lateral side of vertebrae (Figs. 10G, 10H, 11C, and 12A). In proximal to middle arms, a pair of the channels for passages of lateral canals opening inside of ventral furrow, near depression of tube feet entire arms, and distal side of the channels, a pair of the channels for passage of lateral nerves opening (Figs. 10G, 10H, and 11C). They are unrecognizable at distal portion of the arm (Figs. 10G, 10H, 11C, 12A, and 12C).

**Distribution**

Confirmed records of *Astrocladus dofleini*: PHILIPPINES: Cabugan Grande Island, central Philippine, 135 m (*Döderlein, 1927*). JAPAN: Sagami Bay and Tokyo Bay, central-eastern Japan, 2–600 m (*Döderlein, 1911*; *Irimura, 1982*; *Kohtsuka et al., 2017*; this study, type locality, Fig. 1); Toyama Bay, central Japan, 40–80 m (*Fujita & Kohtsuka, 2003*); Tachibana Bay, Nagasaki, western Japan, ca. 40 m (This study, Fig. 1); off Minabe, Shirahama, Wakayama, central Japan, depth unknown (This study, Fig. 1). KOREA: Huksando, southwestern Korea, depth unknown (*Yi & Irimura, 1987*).

**Discussion**

*Döderlein (1902)* described *A. coniferus* and *A. pardalis* on the basis of presence/absence of a large conical tubercle on the distal end of each radial shield. Subsequently, he determined that the presence/absence was an intra-specific character and the synonymized *A. pardalis* with *A. coniferus* (*Döderlein, 1911*).

In our study, although examinations of the lectotype of *A. coniferus* (ZSM 20000897) and the holotype of *A. pardalis* (ZSM 20000898) confirmed these morphological

differences between the two specimens (Figs. 2E and 4C), monophyly of two additional specimens which morphologically agree with the lectotype of *A. coniferus* (NSMT E-13118) and the holotype of *A. pardalis* (NSMT E-13119) was confirmed by our molecular phylogeny (see also "Molecular phylogeny" below). Thus, we follow Döderlein's decision to synonymize these two species.

*Döderlein (1911)* also distinguished *Astrocladus coniferus* and *A. dofleini* as follow: *A. coniferus* possesses conical external ossicles but lacks large tubercles on dorsal surface of proximal portion of arms, whereas *A. dofleini* possesses only granules and large tubercles on the same position of the arms.

*Matsumoto (1917)* made the two species conspecific based on the existence of specimens showing intermediate features (*Matsumoto, 1917*) and *Fujita & Kohtsuka (2003)* followed this classification. *Irimura (1982)* distinguished *A. coniferus* (as "*A. coniferus coniferus*") and *A. dofleini* (as "*A. coniferus* var. *dofleini*") based on presence/absence of large tubercles on dorsal surface of proximal arms. However, *Irimura (1982)* did not recognize any morphological features between the "*A. coniferus* var. *dofleini*" and *A. pardalis* (as "*A. coniferus* var. *pardalis*") except color differences.

In addition to the types of *A. coniferus* and *A. pardalis*, we also studied four paralectotypes (ZSM 20000901/2, ZSM 20000901/3, ZSM20000901/4 and ZMB 5923) of *A. dofleini* and confirmed that the Döderlein's diagnostic characters can not distinguish these species, because:

Ossicles on dorsal surface of proximal portion of arms were granular and conical in *A. coniferus* (Figs. 3A and 4F) and granular in *A. dofleini* (Fig. 8D). Both *A. coniferus* (Fig. 3A) and *A. dofleini* (Fig. 8D) possess large tubercles on dorsal surface of proximal arms.

Instead, *A. coniferus* can be distinguished from *A. dofleini* by the following three morphological characters:

(1) *Shape of ossicles*: Ossicles on periphery of radial shields were conical in *A. coniferus* (Figs. 2F and 4D) but those of *A. dofleini* (Fig. 8C) were granules.

(2) *Shapes of large tubercles*: Large tubercles on dorsal disc were conical in *A. coniferus* (Fig. 2F), whereas those of *A. dofleini* were all domed (Fig. 8C).

(3) Distribution of the large tubercles: Although the large tubercles were only on the peripheral edges of radial shields (Fig. 2F) or absent (Fig. 4C) in *A. coniferus*, those of *A. dofleini* were scattered on the dorsal surface of the disc (Fig. 8C).

These differences were also recognized in other examined materials: 2 specimens of *A. coniferus* (NSMT E-13118 and NSMT E-13119); and 6 specimens of *A. dofleini*. Therefore, we conclude that *A. coniferus* and *A. pardalis* are conspecific and distinct from *A. dofleini*. Our molecular phylogenetic analysis also supports this conclusion (see "Molecular Phylogeny" below).

Additionally, color may also differ in these specimens. On the dorsal side, the 2 examined specimens of *A. coniferus* are vivid orange with yellow patches and arm bands, and the 6 NSMT specimens of *A. dofleini* are uniformly black with small black patches

and arm bands, or yellow with light yellowish small patches and arm bands (Fig. 7). However, we refrain from employing these color variations as diagnostic characters because other color patterns for these species have been recorded (*Irimura, 1982*).

**Astrocladus exiguus** (*Lamarck, 1816*)
(Figs. 13–16)

*Euryale exiguum Lamarck, 1816*, 539.
*Astrophyton exiguum.–Müller & Troschel, 1842*, 125; *Lyman, 1875*, pl. 4, fig. 48; 1882, 257, pl. 47, fig. 1.
*Astrocladus exiguus.–Döderlein, 1911*, 76, 77, 106, 107, pl. 9, fig. 6; 1927, 34, 93, pl. 5 fig. 9; Clark, 1915, 187; Koehler, 1931, 34, 35, pl. 4, figs. 1, 2;h, 60, pl. 1, figs. 1, 2; *Chang, Liao & We, 1962*, 59, 60, pl. 1, figs. 1, 2; Clark & Rowe, 1971, 78, 79, 92, fig. 21; *Cherbonnier & Guille, 1978*, 11, 12, pl. 2, figs. 1, 2; *Baker, 1980*, 63, figs. 28, 33; *Irimura, 1981*, 19; *Liao & Clark, 1995*, 169, 170, fig. 73, pl. 19, fig. 1; *Rowe & Gates, 1995*, 365; *Baker, Okanishi & Pawson, 2018*, 9, 10.
*Gorgonocephalus cornutus Koehler, 1897*, 368, 369, pl. 9, figs. 80, 81; 1899, 73, 74, pl. 12, figs. 95, 96, pl. 13, fig. 98.
*Astrophyton cornutum.—Koehler, 1905*, 127–129, pl. 13, fig. 1, pl. 18, fig. 2.
*Astrocladus dofleini.—Bomford, 1913*, 220, 221, pl. 13, fig. 1 (Non *Astrocladus dofleini*).

**Material examined**
NSMT E-13125, 1 specimen, off Minabe, Wakayama, Japan, depth unknown, 11 November 2012, gill net. NSMT E-13126, 1 specimen, off Minabe, Wakayama, Japan, depth unknown, 4 April 2019, gill net, collected by Sadao Inoue. NSMT E-6265, off Yaku-shima Island, Kagoshima, Japan, 29°47.00′N., 130°22.06′E. 155–170 m, 2 August 2008, 1 m biological dredge, R/V *Soyo-Maru* of Japan Fisheries Research and Education Agency.

**Description of external morphology (NSMT E-13126)**
*Disc*. Disc five-lobed with notched interradial edges, approximately 26 mm in disc diameter. Radial shields and surrounding plates slightly tumid (Fig. 13B). Dorsal disc covered by variously sized conical ossicles, which bear spiny projections on their apices (Figs. 13A and 13B). The larger conical external ossicles separated and scattered, approximately 140–1150 μm in length, having several thorny apical projections (Fig. 13A). Radial shields concealed by ossicles (Figs. 13B and 13C). One large tubercle situated on distal edge of each radial shield, approximately 1.7 mm in diameter (Fig. 13A).

Ventral surface of disc covered by polygonal plate-like ossicles, fully in contact, approximately 170–500 μm in length. Ossicles on ventral plates granule-shaped, approximately 130 μm in length (Fig. 13C). Oral shields, adoral shields, oral plates and ventral arm plates concealed by ossicles (Figs. 13C–13E). Teeth uniformly spiniform, situated on top of dental plates and edges of ventral plates (Fig. 13C). Teeth arranged in

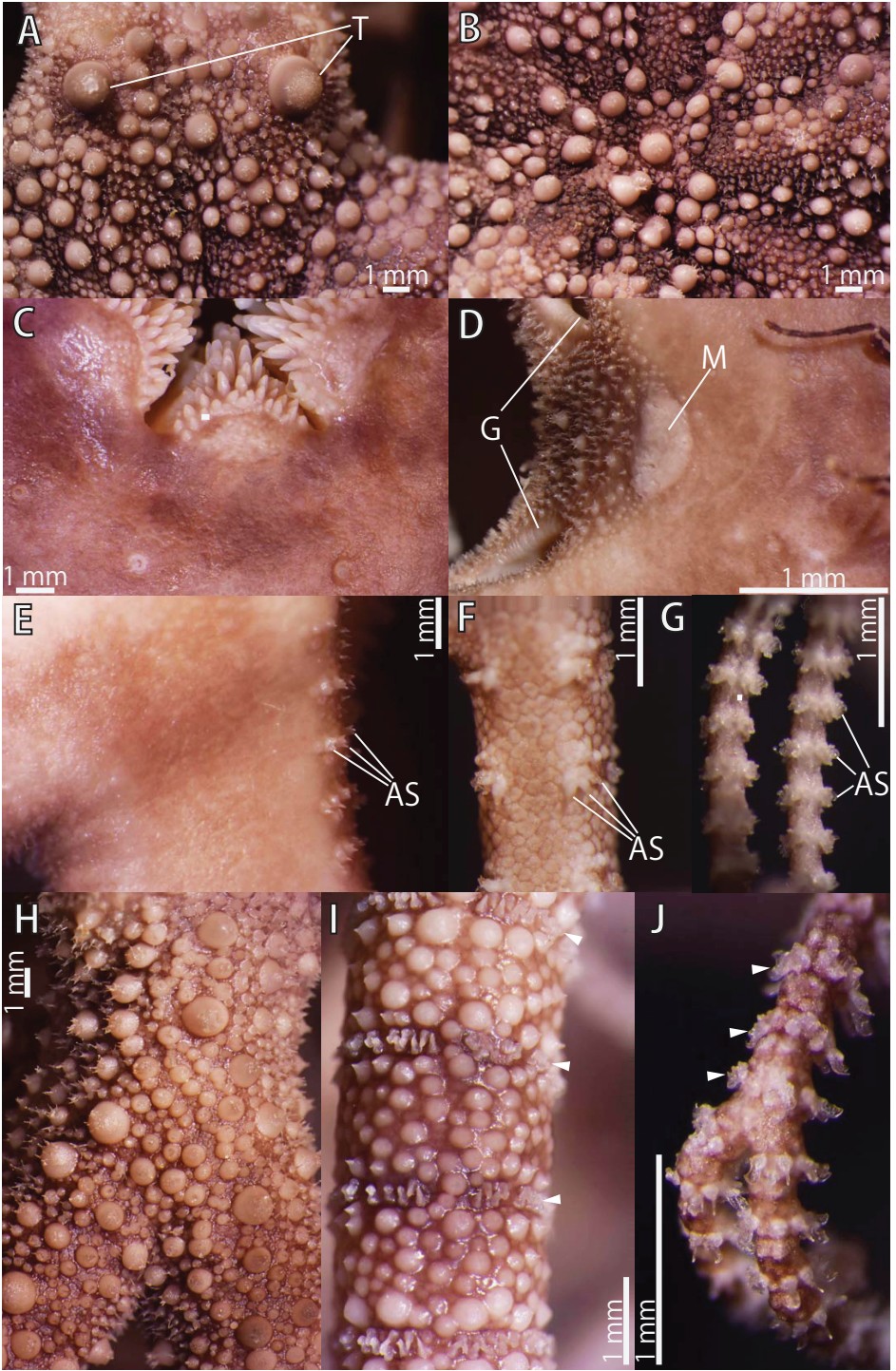

**Figure 13** *Astrocladus exiguus* (NSMT E-13126). (A) Periphery of dorsal disc. (B) Central view of dorsal disc. (C) Ventral disc. (D) Interradial ventral disc. (E–G) Ventral surfaces of arms, proximal (E), middle (F) and distal (G) portion of arm. Dorsal surfaces of arms, proximal (H), middle (I) and distal (J) portion of arm. Arrowheads indicate rows of hooklets on dorsal and lateral side of the arms (I and J). Abbreviations: AS, arm spine; G, genital slit; M, madreporite; T, large tubercle. Photographs of this figure were focus-stacked using the software CombineZM v. 1.0.0 (https://www.softpedia.com/get/Multimedia/Graphic/Graphic-Editors/CombineZM.shtml).

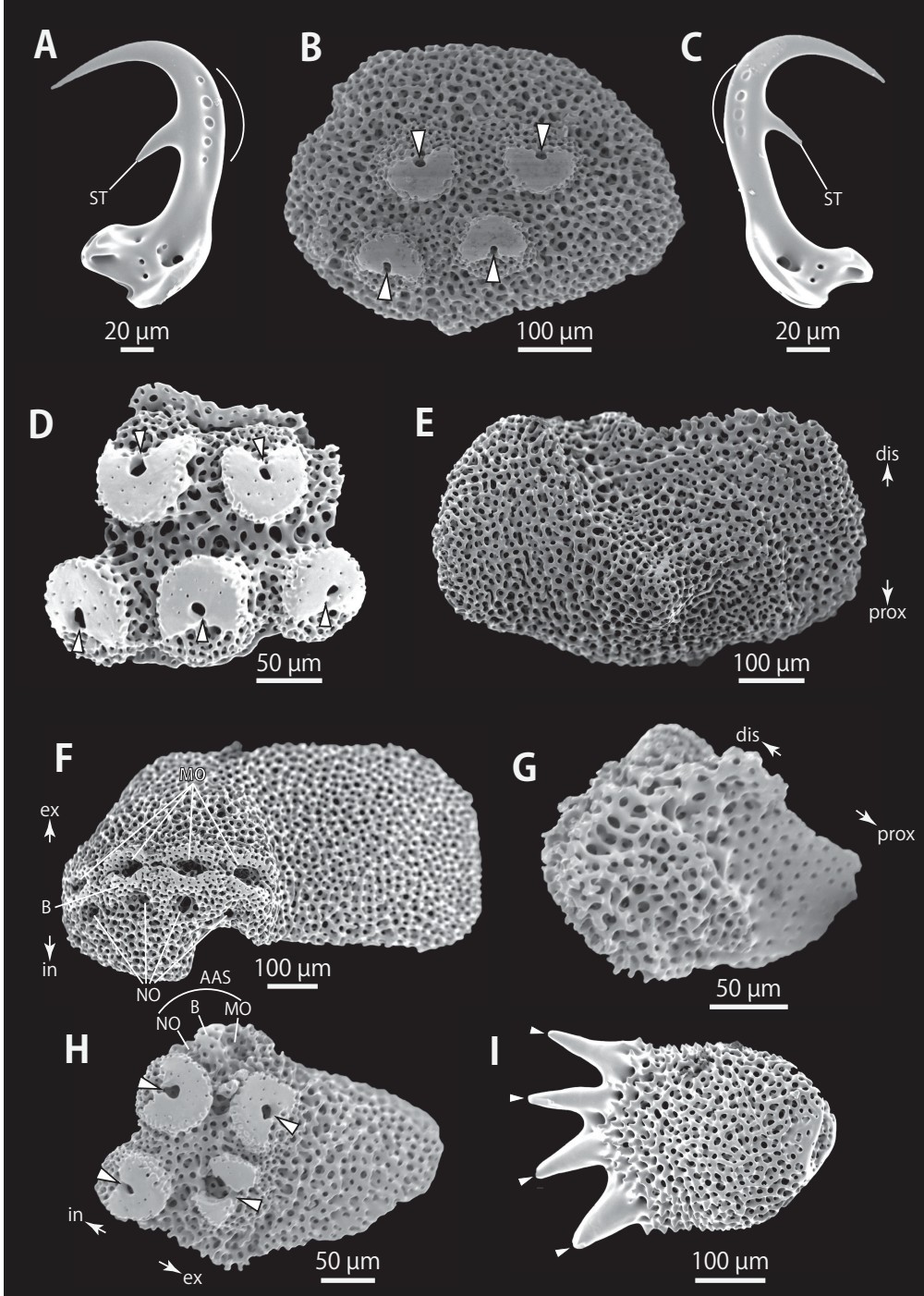

**Figure 14** *Astrocladus exiguus* **(NSMT E-13126). SEM photographs of ossicles.** (A, C) Hooklets on proximal (A) and distal (C) portion of arms, arcs indicate reticular structure. (B and D) Hooket-bearing plate on proximal (B) and distal (D) portion of arm. (E–H) Lateral arm plates on proximal (E and F) and distal (G and H) portion of arms. (I) An arm spine on proximal portion of arm. Arrowheads indicate articulations for hooklets (B, D and H) and terminal projections (I). Orientations: d, dorsal side; dis, distal side; ex, internal side; in, internal side; prox, proximal side; v, ventral side. Abbreviations: AAS, articulation for arm spine; B, border structure; MO, muscle opening; NO, nerve opening; ST, secondary tooth.

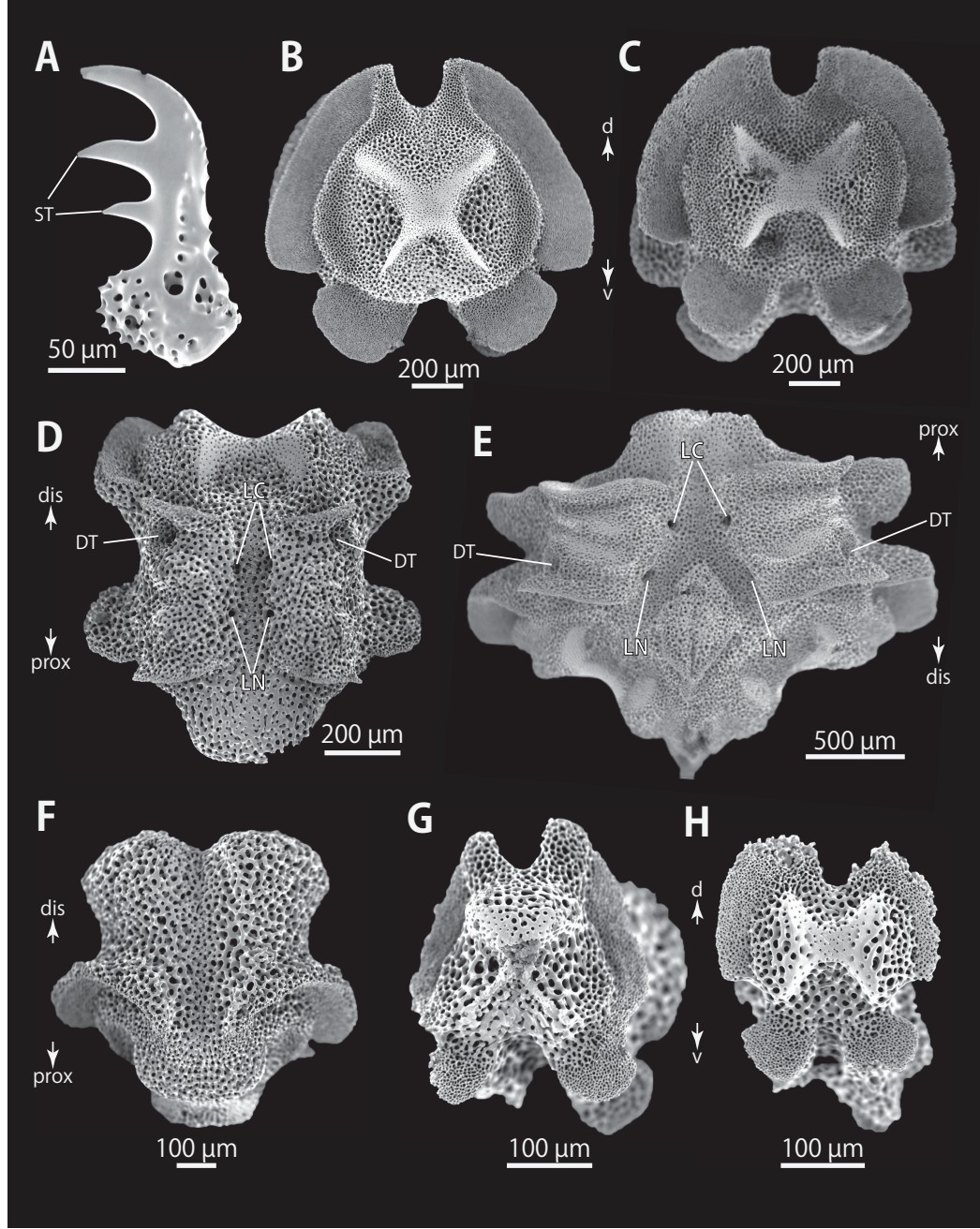

**Figure 15 *Astrocladus exiguus* (MO-2019-19). SEM photographs of ossicles.** (A) An arm spine on distal portion of arm. (B–F) Vertebrae from proximal portion of arm (E is branching vertebra), proximal (B), distal (C), ventral (D and E) and dorsal (F) views. (G and H) Vertebrae from distal portion of arm, proximal (G) and distal (H) views. Orientations: d, dorsal side; dis, distal side; prox, proximal side; v, ventral side. Abbreviations: DT, depression for tentacle; LC, passages of lateral canal; LN, passages of lateral nerve; ST, secondary tooth.                         

1 or 2 transverse rows on ventral plates, approximately 10 in number (Fig. 13C), in a cluster covering ventral-most part of dental plate, approximately 15 in number (Fig. 13C), and in a vertical line, on other areas of dental plates, 2 in number. Size of teeth

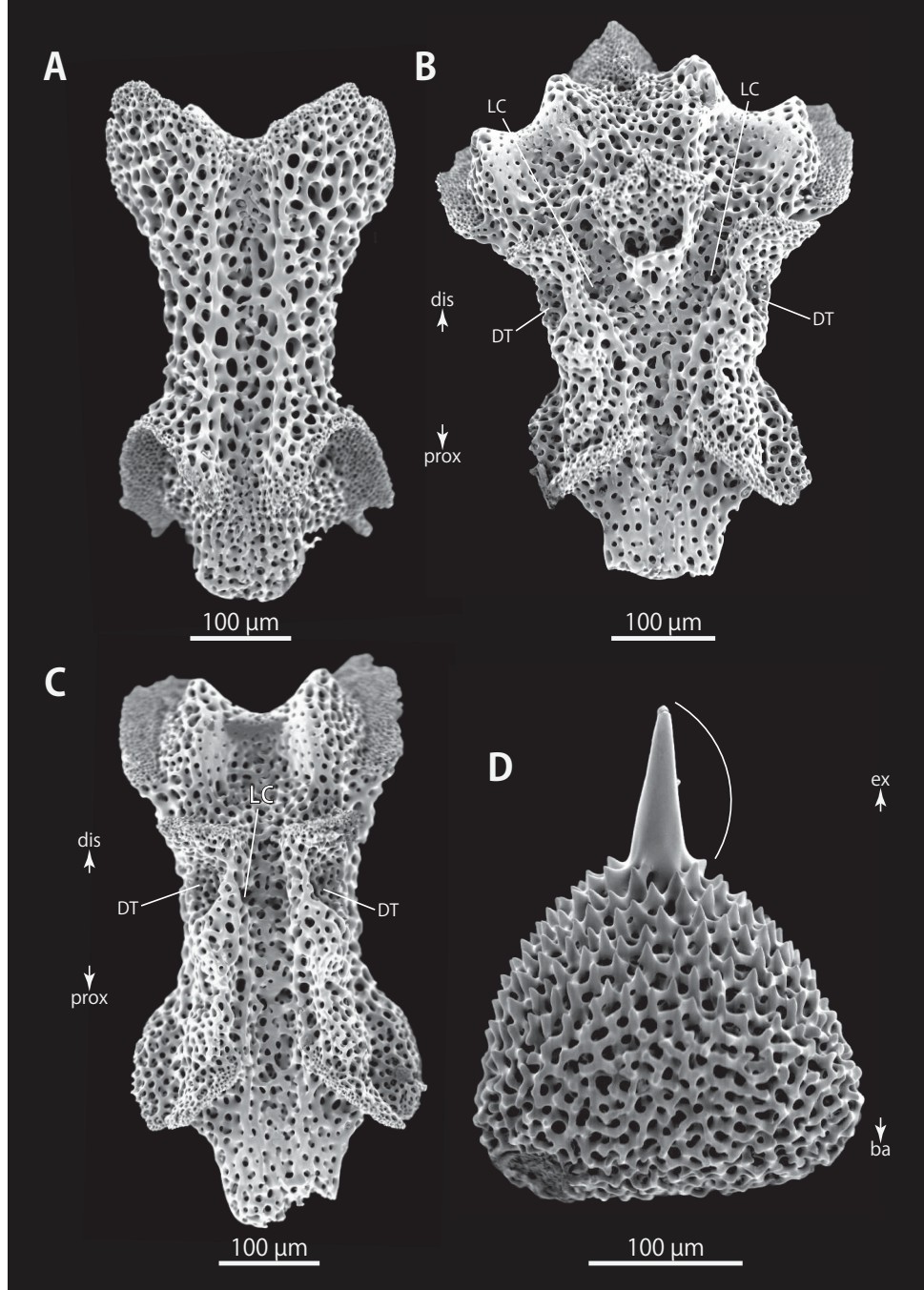

**Figure 16** *Astrocladus exiguus* (NSMT E-13126). SEM photographs of ossicles. (A–C) Vertebrae from distal portion of arm (B is branching vertebra), dorsal (A), ventral (B and C) views. (D) An conical external ossicle on proximal portion of arm, lateral view, an arc indicates a terminal projection. Orientations: ba, basal side; dis, distal side; ex, external side; prox, proximal side. Abbreviations: DT, depression for tentacle; LC, passages for lateral canals.

varying in position on oral and dental plate, approximately 0.3–1 mm in length, 0.3 mm in greatest width on dental plates, and 1 mm in length, approximately 0.2 mm in width on ventral plates (Fig. 13C).

Interradial surface of lateral disc covered by conical ossicles similar to those on dorsal disc (Fig. 13D). They are fully in contact, approximately 40–100 µm in length (Fig. 13D). Two genital slits (0.9 mm long and 0.2 mm wide) in each interradius (Fig. 13D). One large, elliptical madreporite situated on ventral interradius, approximately 0.65 mm in width and 0.35 mm in length (Fig. 13D).

*Arms.* Arms branching. On proximal portion of arm, before first branch, arm 12.0 mm wide and 5.5 mm high, with an arched dorsal surface and flattened ventral surface. Between first branch and second branch, arm width and height abruptly decreasing to 4.3 mm in width and 3.0 mm in height. Subsequently, arms tapering gradually toward arm tip (Figs. 13E–13J).

On dorsal and lateral surface of middle to distal portion of arms, each arm segment covered by single annular, ring-like row of large oblong plates, approximately 700 µm in transverse length (Figs. 13I and 13J). Before third branch, each plate separated by granules. Plates fully in contact from fourth branch and on subsequent distal segments (Fig. 13I). Before third branch, no hooklets (Fig. 13H), after fifth branch, plates appearing and forming an annual band (Fig. 13I). With exception of hooklet-bearing plates, dorsal and lateral surface of arm completely covered by conical, plate-shaped and domed granule-shaped ossicles (Figs. 13H and 13I). Proximal portion of dorsal arm covered by conical ossicles similar to those on dorsal disc, approximately 0.3–1.5 mm in length slightly separated, and plate-shaped external ossicles, fully in contact, approximately 200 µm in length (Fig. 13H). Middle portion of dorsal arm covered by domed granules, approximately 170–220 µm in length, and plate-shaped ossicles, approximately 110 µm in length (Fig. 13I). The larger conical ossicles sometimes carry spiny projections. Distal portion of dorsal arm covered by granule-shaped external ossicles, approximately 50 µm in length (Fig. 13J). In proximal to middle portion of arms, ventral surface covered by polygonal and plate-shaped ossicles, similar to those on ventral disc, fully in contact, approximately 150–250 µm in length at proximal region, and 60–260 in length distally (Fig. 13H and 13I). Distal portion of ventral arm covered by granule-shaped external ossicles, slightly in contact, approximately 40 µm in length (Fig. 13J). With exception of the articulations with arm spines and/or hooklets, lateral arm plates and ventral arm plates concealed by skin and ossicles (Figs. 13E–13G). First to fifth tentacle pore without single spine; sixth pores with 1 spine, seventh and subsequent pore with 2 or 3 spines (Fig. 13E). Distally, the number of arm spines decrease gradually to 2 toward arm tip (Fig. 13G). Arm spines approximately one-third to one-fourth of length (ca. 25–35%) of corresponding arm segment, covered by thin integument (Figs. 13E–13G).

*Color.* Dorsal surface dark brown with whitish patches on disc and bands on arms. Ventral surface whitish but slightly brown on disc.

*Ossicle morphology of NSMT E-13126*: Each hooklet with single inner tooth and reticular structure (Figs. 14A and 14C). Hooklet-bearing plates with 4 tubercle-shaped articulations

for hooklets in proximal portion of the arm (Fig. 14B), approximately 5 articulations in distal portion (Fig. 14D); articulations forming 2 parallel rows (Figs. 14B and 14D). Lateral arm plates long, both distal and proximal edges straight (Fig. 14E). On proximal portion of arm, lateral arm plates without perforation-like structures on dorsal side and pairs of simple nerve and muscle openings on ventral-external side (Figs. 14E and 14F) and on distal portion of arms, no perforation-like structure on dorsal side and a pair of nerve and muscle openings of articulation for arm spine beside boarder structure and 4 articulations for hooklets on ventral surfaces (Figs. 14G and 14H). Arm spines in proximal portion of arm ovoid, with four small projections, approximately one-third the height of the height of spine (Fig. 14I). In distal portion, arm spines transformed into hooks with 2 inner secondary teeth, respectively (Fig. 15A).

All vertebrae with hourglass-shaped streptospondylous articulations (Figs. 15B, 15C, 15G, and 15H), and distal side of branching vertebra slightly wider than in non-branching vertebra and with 2 articulation surfaces (Figs. 15E and 16B). Surfaces of lateral furrows smooth, with no special ornamentation (Figs. 15F and 16A). Depressions for tube feet openings in distal part of ventral-lateral side of vertebrae (Figs. 15D, 15E, and 16C). Two pairs of the channels for passages of lateral canals and lateral nerves opening on ventral groove of vertebrae in proximal portion of arm (Figs. 15D and 15E). In distal portion of arm, only radial water canal observed (Fig. 16B). External ossicles on dorsal periphery of radial shields conical, approximately 150 μm in length and 200 μm in height with a spiny apical projection, approximately 100 μm in length (Fig. 16D).

**Distribution**. Widely distributed in Indo-Western Pacific Ocean. Depth range 18–494 m.

**Discussion**

*Astrocladus exiguus* can be distinguished from other congeners by its covering of large tubercles and ossicles on dorsal surface of disc and proximal regions of arms: large tubercles are conical and scattered; ossicles conical with acute thorny tips (Table 2). Our molecular phylogeny showed that the two examined specimens of *A. exiguus* were monophyletic and distinguished from *A. coniferus* and *A. dofleini* (See "Molecular Phylogeny" below).

**Astrocladus annulatus** *Matsumoto, 1912a*
(Fig. 17)

   *Astrophyton annulatum Matsumoto, 1912a*: 206, figs. 17–18.
   *Astrocladus annulatus.—Matsumoto, 1912b*: 389; 1915: 56–57; 1917: 75–77, fig. 22; Clark, H. L., 1915: 187; *Irimura, 1981*: 19; *Irimura & Kubodera, 1998*: 138.

**Type material examined**
The holotype (UMUTZ-Oph-26): Off Misaki, Miura, Sagami Bay, Kanagawa, Japan, depth and collected date unknown, disc cut into two halves, probably done by Hikoshichiro Matsumoto (*Fujita, 2006*).

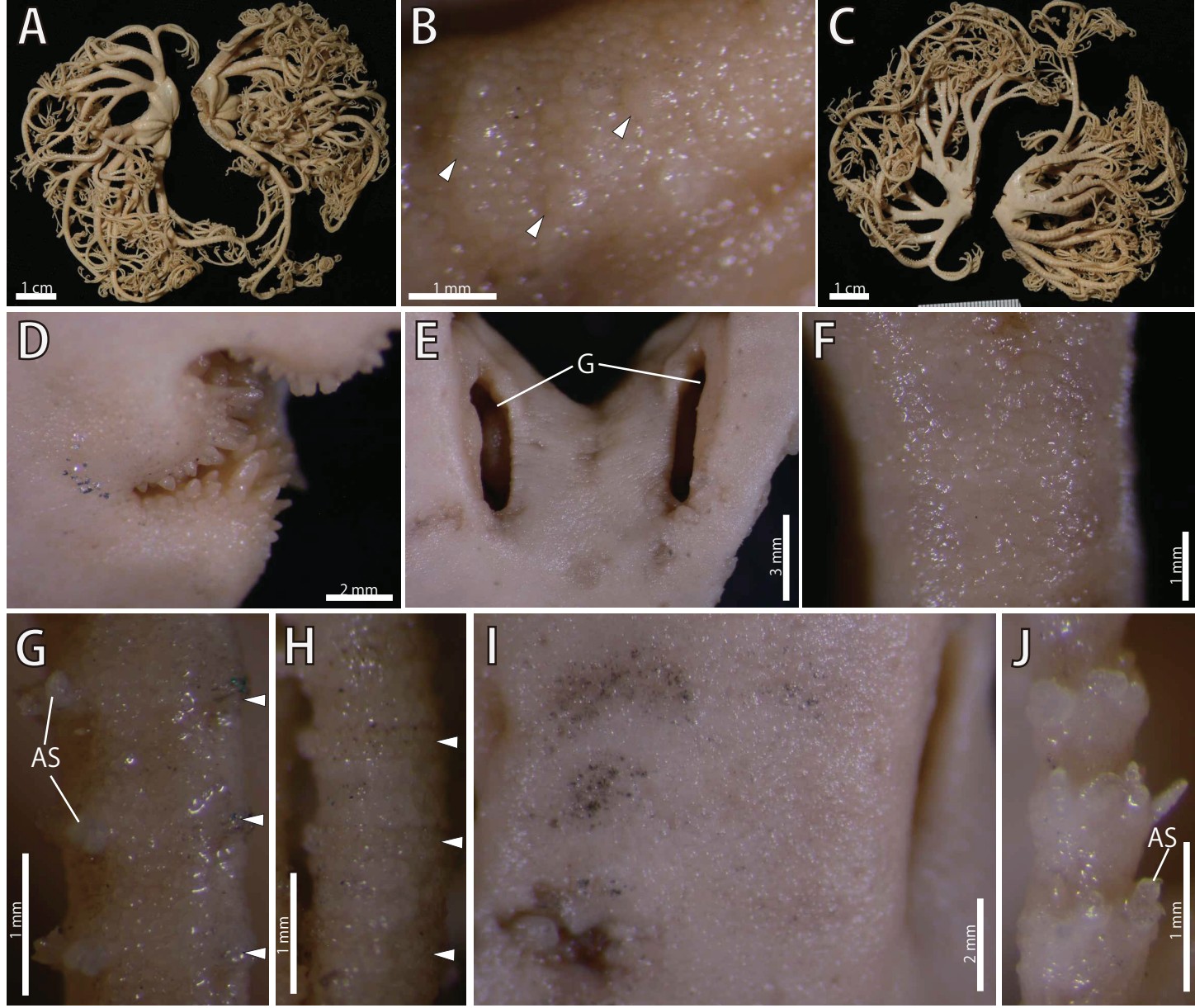

**Figure 17** *Astrocladus annulatus*, **holotype (UMUTZ-Ophi-26).** (A) Dorsal view. (B) Dorsal surface of periphery disc (C) Ventral view. (D) Jaws. (E) Interradial ventral disc. (F–H) Dorsal surface of arms, proximal (F), middle (G) and distal (H) portion of arm. (I and J) Ventral surface of arms, proximal (I) and distal (J) portion of arm. Arrowheads indicate rows of hooklets on dorsal and lateral side of the arms (B, G and H). Abbreviations: AS, arm spine; G, genital slit.

**Description of holotype (UMUTZ-Ophi-26)**

*Disc.* Disc five-lobed with notched interradial edges, 22 mm in diameter. Dorsal disc covered by granules, approximately 140–280 µm in length (Fig. 17B) Radial shields and their surrounds tumid, concealed by ossicles (Fig. 17A), approximately 1.1 mm in length, almost reaching disc center (Fig. 17A). Large domed tubercles, approximately 450 µm in length scattered on radial shields (Fig. 17B).
Ventral surface of disc covered by polygonal plate-like ossicles, fully in contact, approximately 160–200 μm in length (Fig. 17D). Oral shields, adoral shields, oral plates and ventral arm plates concealed by ossicles (Fig. 17D). Teeth uniformly spiniform, on top of dental plates and edges of ventral plates (Fig. 17D). Teeth approximately 8, arranged in 1 or 2 transverse rows on ventral plates in a cluster covering ventral-most part of dental plate, approximately 10 in number (Fig. 17D). Size of teeth variable, approximately 1 mm in greatest length on dental plates, approximately 0.5 mm on oral plates (Fig. 17D). Interradial surface of lateral disc covered by thick skin (Fig. 17E). Two genital slits (4.5 mm long and 1 mm wide) in each interradius (Fig. 17E). One small, elliptical madreporite on ventral interradius.

*Arms.* Arms branching. On the proximal portion, before first branch, arm 4.3 mm wide and 3.5 mm high, with an arched dorsal surface and flattened ventral surface (Figs. 17A and 17C). Between first and second branch, arm width and height abruptly decreasing to 3 mm in width and 1.8 mm in height. Subsequently, arms tapering gradually toward arm tip (Figs. 17A and 17C).

On dorsal and lateral surface, each arm segment covered by single annular row of large oblong plates (Figs. 17G and 17H). With exception of hooklet-bearing plates, dorsal and lateral surface of arm completely covered by polygonal plate-like ossicles, approximately 170–290 in length at proximal portion of arms, and subsequently decreasing in size to arm tip (Figs. 17G and 17H). Ventral side of arms covered by skin which completely conceals the external ossicles, lateral arm plates and ventral arm plates, with exception of the part where lateral arm plates and arm spines articulating (Figs. 17I and 17J). Tentacle pores without arm spines before first branch; 3 or 4 spines after second branch. Distally, number of arm spines decrease gradually to 1 towards arm tip (Fig. 17J). In proximal portion, arm spines approximately one-fourth to one-fifth of length of corresponding arm segment, and covered by thin integument; subsequently relative length increase, exceeding half length of corresponding arm segment on distal portion of arm (Fig. 17J).

*Color.* Uniformly creamy white (Fig. 17).

**Distribution**. JAPAN: Sagami Sea, Off Misaki, Kanagawa, central-eastern Japan depth unknown (*Matsumoto, 1917*); Seto, Wakayama, central Japan, depth unknown (*Irimura, 1981*); East China Sea, western Japan, 200 m (*Irimura & Kubodera, 1998*).

**Discussion**

*Astrocladus annulatus* was originally described by *Matsumoto (1912a)* based on the holotype collected from off Misaki, Sagami Bay. It has never been re-collected from the type locality and never re-described so far. In our examination of the holotype, we confirmed the diagnostic character of this species, namely granules on dorsal surface of body (Fig. 17B) and continuous hooklet-bearing plates on proximal portions of arms (*Matsumoto, 1912a*; Table 2).

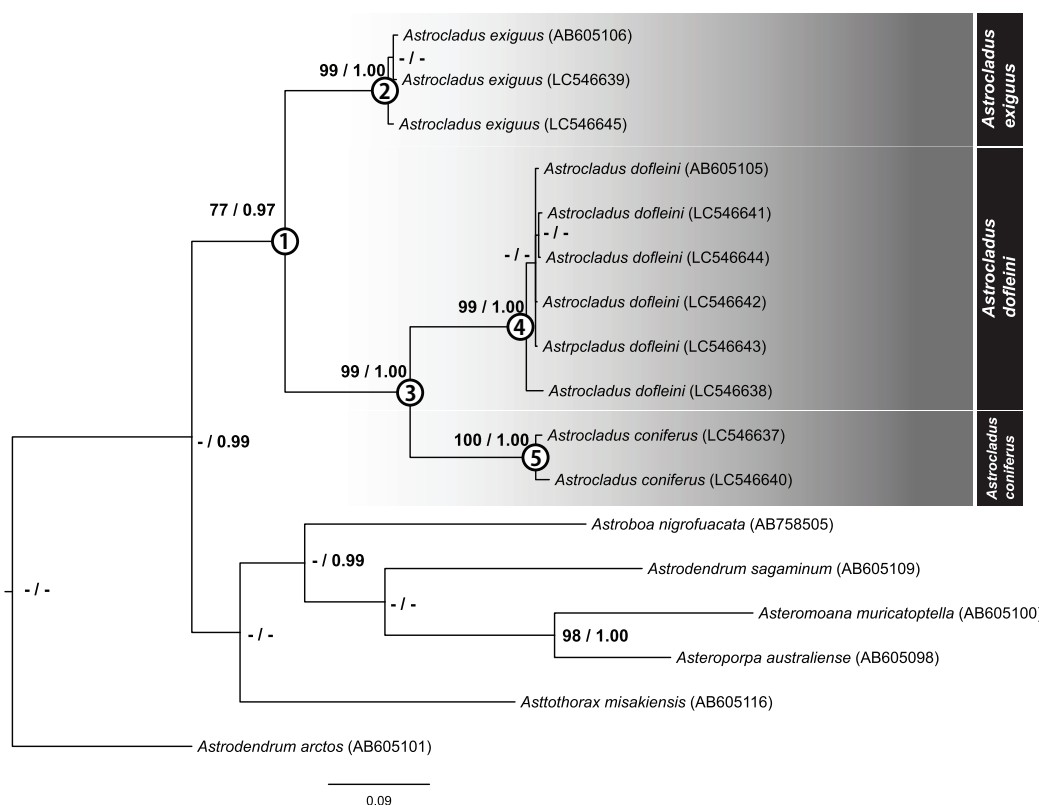

0.09

**Figure 18 Maximum likelihood tree based on a partial sequence of mitochondrial COI gene (699 bp).** Support values for each node are shown by maximum likelihood bootstrap values (%) and Bayesian posterior probabilities. Numerals (1–5) in circles at nodes refer to the clade number discussed in the text. Bootstrap value less than 74% and Bayesian posterior probability value less than 0.97 and for each node were shown by as "–".

## MOLECULAR PHYLOGENY

### Phylogenetic tree and assignation of species to each detected clade

After removal of ambiguous aligned sites, 699 bp of COI were obtained for 10 specimens. The ML tree of concatenated sequence is shown in Fig. 18. The Bayesian tree also showed the same topology. In the ML analyses, monophyly of the genus *Astrocladus* was weakly supported (Fig. 18, Clade 1, bootstrap 77%, BPP 0.97). Within this clade, two clades (Fig. 18, Clade 2, bootstrap 99%, BPP 1.00; Clade 3, bootstrap 99%, BPP 1.00) were detected. The clade 3 was subdivided into two clades (Fig. 18, Clade 4, bootstrap 99%, BPP 1.00; Clade 5, bootstrap 99%, 0.99).

The specimens used in each clade were found to be identified as *A. exiguus* (Clade 2), *A. dofleini* (Clade 4) and *A. coniferus* (Clade 5), respectively (See also remarks of *A. dofleini* and *A. exiguus* above).

### Genetic distances

Mean genetic distances within each clade were 0.67% in *A. exiguus* (Clade 2, 3 specimens), 0.7% in *A. dofleini* (Clade 4, 3 specimens) and 1.3% in *A. coniferus* (Clade 5, 3 specimens). Genetic distances were 13% between *A. dofleini* and *A. coniferus*, 14.7%

between *A. dofleini* and *A. exiguus*, and 14.6% between *A. coniferus* and *A. exiguus*, respectively. Intra-clade distance (0.67 to 1.3%) was about ten folds smaller than inter-clade distance (13 to 14.7%).

## DISCUSSION

Our molecular phylogenetic analyses suggest that *A. exiguus*, *A. coniferus* and *A. dofleini* should be assigned to separate taxa. Genetic distance analysis showed that the inter-clade distances exceed intra-clade values. In previous studies of ophiuroids, genetic distance corresponding to species differences range from approximately 2.2–23% (*Okanishi & Fujita, 2018*). Therefore, the distances between current clades (13–14.7%) are within this range.

In our analysis, we found that *A. coniferus* and *A. dofleini* form a clade (Clade 3). Therefore, a possible classification would be to unite *A. coniferus* and *A. dofleini* as the same species (*A. coniferus*) and subdivide *A. coniferus coniferus* and *A. coniferus dofleini* under *A. coniferus*, as has been done in the past (*Fedotov, 1926*; *Irimura, 1982*). However, since the genetic distance between *A. dofleini* and *A. coniferus* is comparable to the distance of the two species from *A. exiguus*, which is considered to be a separate species in terms of morphology, *A. dofleini* and *A. coniferus* are herein shown to be separate species.

## CONCLUSIONS

In the present study, morphological observations of type and non-type specimens revealed that *Astrocladus pardalis* (*Döderlein, 1902*) is a junior synonym of *A. coniferus* (*Döderlein, 1902*). Morphological observations and molecular phylogenetic analysis revealed that *A. coniferus* and *A. dofleini* (*Döderlein, 1910*) are different species. Therefore, 4 species, *A. annulatus*, *A. coniferus*, *A. dofleini* and *A. exiguus* occurin Japan. Additional molecular analyses including *A. annulatus* and examination of type specimens of *A. exiguus* are required to finally clarify the taxonomy of Japanese basket stars of the genus *Astrocladus*.

## ACKNOWLEDGEMENTS

We are most grateful to David L. Pawson of National Museum of Natural History, Smithsonian Institution for his critical reading of the manuscript and constructive comments, and to Bernhard Rüthensteiner of the Zoologische Staatssammlung München, Peter Bartsch of Museum für Naturkunde der Humboldt-Universität zu Berlin and to Rei Ueshima The University Museum, The University of Tokyo for their help in observation of specimens. Thanks are also extended to the captains and crew members of the R/V *Soyo-Maru* of Japan Fisheries Research and Education Agency, and colleagues on board for their assistance in collecting the specimens, and Tetsuya Kato and Keita Harada of Kyoto University Aquarium, their providing specimens.

### Funding

This work was supported by a grant from the Research Institute of Marine Invertebrates, KAKENHI Grant Numbers 09041155, 125750008, 22570104, 25440226, 17K07549, and

by the integrated research, "Geological, biological, and anthropological histories in relation to the Kuroshio Current" and "Spatiotemporal Analyses on Origins and Properties of the Biodiversity Hotspots in Japan", conducted by the National Museum of Nature and Science. There was no additional external funding received for this study. The funders had no role in study design, data collection and analysis, decision to publish, or preparation of the manuscript.

## Grant Disclosures

The following grant information was disclosed by the authors:
Research Institute of Marine Invertebrates, KAKENHI: 09041155, 125750008, 22570104, 25440226 and 17K07549.

## Competing Interests

The authors declare that they have no competing interests.

## Author Contributions

- Masanori Okanishi conceived and designed the experiments, performed the experiments, analyzed the data, prepared figures and/or tables, authored or reviewed drafts of the paper, and approved the final draft.
- Hisanori Kohtsuka performed the experiments, prepared figures and/or tables, collected examined specimen, and approved the final draft.
- Toshihiko Fujita conceived and designed the experiments, performed the experiments, authored or reviewed drafts of the paper, and approved the final draft.

## Data Availability

The raw sequence data of COI genes are available in the Supplemental Files and at GenBank: LC546637–LC546645.

The museum catalog numbers (NSMT E-13118-13126) of the specimens examined are described in Table 1.

## Supplemental Information

Supplemental information for this article can be found online at http://dx.doi.org/10.7717/peerj.9836#supplemental-information.

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
