# Peer review of "A taxonomic review of the genus Astrocladus (Echinodermata, Ophiuroidea, Euryalida, Gorgonocephalidae) from Japanese coastal waters"

_PeerJ, doi:10.7717/peerj.9836_

## Round 0.1 · original submission · Minor Revisions

Both reviewers were complimentary and felt that your work was carefully and meticulously conducted. They each raised some issues that, when addressed will result in a manuscript that is ready for publication. I trust that you will make these corrections/clarifications and return the manuscript. I do apologize for the time it took to reach this decision but during this COVID-19 crisis, it has taken some reviewers longer than anticipated to be able to complete their reviews. I hope you understand.

·

Basic reporting

See attached files. review below. MS is basically good. But requires editing.

Experimental design

Acceptable and proficient.

Validity of the findings

Yes.

Additional comments

Overview: Okanishi et al. address the species of Astrocladus present in Japan. Three are addressed, but morphological and molecular data show that there are in fact four species.

A solid, well-researched and meticulously crafted piece of work.

******I thought a key to Japanese species OR a table summarizing characters between the four Japanese species would be a distinct improvement. ********

Other comments:

*The MS displays some difficulty in summarizing results and explaining the results of their work.

One specific issue: the abstract and the introduction assumes that Astrocladus coniferus has two synonyms: A. pardalis and A. dolfleini.. and yet, WoRMS lists both of these species as “accepted”. This confused me until I read further… While this could simply be a shortcoming of the WoRMS editor, the authors cannot ignore this..and must cite the proper citation to support their argument.

English is not the primary language of the authors, and although they are clearly proficient in the taxonomic descriptions and technical sections there are places where the language regarding synonymies and designation of lectotypes is a bit difficult to follow.

Edits are made directly to the pdf.

·

Basic reporting

The article has been proof read by an English speaking taxonomist and so in general is well written and professional. I have made some small alterations in the attached pdf.

The illustrations are numerous, clear and useful.

The cited literature needs a little attention, there is a mixture of abbreviated and full journal titles, some missing data, etc.

Experimental design

Not really relevant. The authors have accessed all the specimens they could of the genus from Japanese waters. The species are relatively rare.

I have checked the COI sequences provided and blasted them against my own reference database. I agree with the phylogeny provided in the manuscript.

Validity of the findings

I agree with the taxonomic decisions made in the manuscript. They reestablish an old species that had been previously synonymsied.

Additional comments

I have only two general comments and some minor editorial suggestions that are in the attached annotated pdf.

1. Some of the material examined have a voucher code starting with MO. What is the collection status of these specimens? Will they be placed in a museum (eg in the NSMT) after the study is published? If so it would be better if you put NSMT registration numbers in the manuscript.

2. I am a little confused by the distal hooklet-bearing plates that you illustrate (pl 6A, 9J, 14H). Are these really lateral arm plates with arm spines and hooklets, or are they distal hooklet bearing plates that lie on top of the lateral arm plates. Your descriptions and figure captions do not make this clear. In fact you say that the lateral arm plates are completely covered in granules/plates (eg lines 388-9). If they are completely covered, how do the arm spines stick out? This needs to be made more clear.

---

## Round 0.2 · accepted · Accept

Thank you for your careful response to the comments from two reviewers. I am satisfied that your revised manuscript satisfactorily addresses all of the points raised in the initial review.